# Recent Advances in Cardiac Tissue Engineering for the Management of Myocardium Infarction

**DOI:** 10.3390/cells10102538

**Published:** 2021-09-25

**Authors:** Vineeta Sharma, Sanat Kumar Dash, Kavitha Govarthanan, Rekha Gahtori, Nidhi Negi, Mahmood Barani, Richa Tomar, Sudip Chakraborty, Santosh Mathapati, Dillip Kumar Bishi, Poonam Negi, Kamal Dua, Sachin Kumar Singh, Rohit Gundamaraju, Abhijit Dey, Janne Ruokolainen, Vijay Kumar Thakur, Kavindra Kumar Kesari, Niraj Kumar Jha, Piyush Kumar Gupta, Shreesh Ojha

**Affiliations:** 1Stem Cell and Molecular Biology Laboratory, Department of Biotechnology, Indian Institute of Technology Madras, Bhupat and Jyoti Mehta School of Biosciences, Chennai 600036, India; vineetabt17@gmail.com (V.S.); govarthanan_kavitha@yahoo.co.in (K.G.); 2Heat Transfer and Thermal Power Laboratory, Department of Mechanical Engineering, Indian Institute of Technology Madras, Chennai 600036, India; 12345sanat@gmail.com; 3Department of Biotechnology, Sir J. C. Bose Technical Campus, Kumaun University, Nainital 263136, India; rekhagahtori11@gmail.com; 4Department of Chemistry, DSB Campus, Kumaun University, Nainital 263001, India; nidhi.negi2@gmail.com; 5Medical Mycology and Bacteriology Research Center, Kerman University of Medical Sciences, Kerman 7616913555, Iran; mahmoodbarani7@gmail.com; 6Department of Chemistry and Biochemistry, School of Basic Sciences and Research, Sharda University, Knowledge Park III, Greater Noida 201310, India; richa.tomar@sharda.ac.in; 7School of Chemistry, University of New South Wales, Anzac Parade, Kensington, NSW 2033, Australia; sudipchakraborty3@gmail.com; 8Translational Health Science and Technology Institute, NCR Biotech Science Cluster, 3rd Milestone, Faridabad-Gurugram Expressway, Faridabad 121001, India; santosh@thsti.res.in; 9Department of Biotechnology, Rama Devi Women’s University, Bhubaneswar 751022, India; dillipkumar.bishi@gmail.com; 10School of Pharmaceutical Sciences, Shoolini University of Biotechnology and Management Sciences, Solan 173212, India; poonam.546@shooliniuniversity.com; 11Discipline of Pharmacy, Graduate School of Health, University of Technology Sydney, Ultimo, Sydney, NSW 2007, Australia; Kamal.Dua@uts.edu.au; 12Australian Research Centre in Complementary and Integrative Medicine, Faculty of Health, University of Technology Sydney, Ultimo, Sydney, NSW 2007, Australia; 13School of Pharmaceutical Sciences, Lovely Professional University, Phagwara 144001, India; singhsachin23@gmail.com; 14ER Stress and Mucosal Immunology Laboratory, School of Health Sciences, University of Tasmania, Launceston, TAS 7248, Australia; rohit.gundamaraju@utas.edu.au; 15Department of Life Sciences, Presidency University, College Street, Kolkata 700073, India; abhijit.dbs@presiuniv.ac.in; 16Department of Applied Physics, School of Science, Aalto University, 00076 Espoo, Finland; janne.ruokolainen@aalto.fi (J.R.); kavindra.kesari@aalto.fi or; 17Biorefining and Advanced Materials Research Centre, Scotland’s Rural College (SRUC), Kings Buildings, Edinburgh EH9 3JG, UK; Vijay.Thakur@sruc.ac.uk; 18School of Engineering, University of Petroleum and Energy Studies (UPES), Dehradun 248007, India; 19Department of Bioproducts and Biosystems, School of Chemical Engineering, Aalto University, 00076 Espoo, Finland; 20Department of Biotechnology, School of Engineering and Technology, Sharda University, Knowledge Park III, Greater Noida 201310, India; nirajkumarjha2011@gmail.com; 21Department of Life Sciences, School of Basic Sciences and Research, Sharda University, Knowledge Park III, Greater Noida 201310, India; 22Department of Pharmacology and Therapeutics, College of Medicine and Health Sciences, United Arab Emirates University, Al Ain P.O. Box 17666, United Arab Emirates

**Keywords:** myocardial infarction, stem cells, regeneration, biomaterial, cardiomyocytes, tissue engineering

## Abstract

Myocardium Infarction (MI) is one of the foremost cardiovascular diseases (CVDs) causing death worldwide, and its case numbers are expected to continuously increase in the coming years. Pharmacological interventions have not been at the forefront in ameliorating MI-related morbidity and mortality. Stem cell-based tissue engineering approaches have been extensively explored for their regenerative potential in the infarcted myocardium. Recent studies on microfluidic devices employing stem cells under laboratory set-up have revealed meticulous events pertaining to the pathophysiology of MI occurring at the infarcted site. This discovery also underpins the appropriate conditions in the niche for differentiating stem cells into mature cardiomyocyte-like cells and leads to engineering of the scaffold via mimicking of native cardiac physiological conditions. However, the mode of stem cell-loaded engineered scaffolds delivered to the site of infarction is still a challenging mission, and yet to be translated to the clinical setting. In this review, we have elucidated the various strategies developed using a hydrogel-based system both as encapsulated stem cells and as biocompatible patches loaded with cells and applied at the site of infarction.

## 1. Introduction

Cardiovascular disease, predominantly MI, is attributed the highest mortality rate worldwide [1]. Reduced contractility and function, irregular left ventricle remodeling, and uneven stress distribution in the heart muscle are among the complications occurring post-MI, eventually resulting in catastrophic heart failure. According to the American Heart Association (AHA)’s “Heart Disease and Stroke Statistics—2021”, the prevalence of CVD (including heart failure, hypertension, and stroke) in the US population is 49.2% in the age range of 20 years and above [2]. In 2014, 150,000 people died due to MI; thus an estimated approximately 14% of global death occurs mainly due to MI. Furthermore, MI survivors are also 15 times more likely to develop post-disease complications that lead to heart failure, and are prone to die sooner rather than later compared to the normal population [1]. Cardiac ischemia-related deaths have also ascended to the top of the list of causes of death in India, the United States, and Europe, apart from MI [1,3,4]. After MI incidence, male and female patients above 45 years of age have a lower life expectancy, of 8.2 and 5.5 years, respectively [1]. Socioeconomic burdens such as health care infrastructure and treatment costs (USD 11.5 billion) have made MI one of the top ten most expensive illnesses in the United States [1,5]. Researchers and clinicians around the world have been working extensively to reduce the global incidence of MI and develop significant cost-effective treatment strategies to reduce the mortality rate from MI.

The human heart is a complex organ composed of various types of cells such as cardiomyocytes (CM), fibroblasts, endothelial cells, valve interstitial cells, and resident cardiac stem cells. The cells of the heart are very active metabolically, as it physiologically requires adenosine tri-phosphate (ATP) for its function. Nonetheless, the heart lacks endogenous repair or regeneration potential, thus it remains devoid of regenerative capacity. Any defect in size or deficiency in cardiomyocyte numbers leads to life-threatening MI-related cardiovascular complications [6]. Currently, mitigation of CVDs by pharmaceutical drugs and other clinical practices have effectively improved the patient’s survival and quality of life after tissue damage [7]. However, this remains only a short-term solution of temporary duration; the permanent curative would be via heart transplant. Severe shortage of donor organs, post-graft complications, and the limited efficacy of pharmacological interventions has placed the emphasis on cell-loaded scaffold-based therapeutic approaches for cardiovascular complications (CVDs).

The emergence of cardiac tissue engineering (CTE) has not only given substantial hope for resolving or rescuing the damaged heart after MI but also for prompting the regeneration of the damaged myocardium, thus providing a permanent curative. The idea of CTE was first impelled in 1995 by in vitro-generated cardiac tissue obtained from embryonic chicken CMs. This further ushered in the prospect of new research areas around CTE, mainly idealised to translate the bench to bedside. CTE primarily aims to recapitulate the in vivo cardiac niche under in vitro conditions. Therefore, the long-term goals of CTE are considered the construction of in vitro-fabricated tissues for in vivo cardiac repair and regeneration, in vitro preclinical models for evaluation of drug toxicity, and disease models for understanding the development and pathophysiology of heart-related disorders [8]. With the global rise in CVD cases, it is essential to reinforce the treatment modalities for better disease management. In the current scenario, the previously mentioned CTE is considered as at the forefront; however, the conducting of numerous clinical trials is crucial in prioritizing CTE in clinical practice. 

Delivery of an engineered scaffold loaded with stem cells, CM mitogens, or pharmacological molecules directly to the infarcted site via either the intra-coronary or intra-myocardial mode leads to the relatively prompt recovery of the infarcted tissue, followed by regeneration and regaining of functional significance. Still to be addressed are the current roadblocks to successive clinical utilization, such as an optimized protocol for stem cell-derived CMs and their source, biomaterials for CM cultures, as well as their delivery strategies [9]. Advanced delivery approaches using injectable or patch-based methods are recently gaining significant attention due to their complexity in design and versatility in application. The most advanced technology, using iPSC-derived CM-loaded microfluidic devices, has now been providing unprecedented opportunities to understand the mechanisms of MI development. This technology can also be employed to study the effects of drugs in the preclinical drug screening phase [10].

In this review, we elaborate the recent technological advances in cardiac tissue engineering, particularly the therapeutic approach to regenerating the infarcted myocardium. This review emphasizes mainly cell-based therapy, patch-based therapy, and microfluidics for studying micro-tissue physiology under laboratory set-up. The overall representation of the aforesaid approaches has been elucidated in Figure 1.

## 2. Regenerative Therapy

Heart failure due to the progressive complications of MI mainly occurs due to the limited intrinsic regenerative potential of the myocardium. A left-ventricular assist device (LVAD), fixed internally to relieve the pressure on the heart left region, is the only available medication for the management of post-MI complications. This temporarily delays post MI complications; meanwhile, in such cases, a heart transplant can be the only permanent solution. However, the lack of organ donors has led researchers and clinicians to contemplate alternative therapies available immediately after MI for minimizing cardiomyocyte damage and thereby preventing subsequent heart failure. Under these conditions, cell-based therapy is ideal for repairing the initial injury, restoring lost cardiomyocytes, and preventing the development of a scar (which impairs cardiac function) (Figure 2). Many research groups are currently investigating the possibility of restoring cardiac function by replacing lost cardiomyocytes or via rejuvenating the resident cardiac stem cell population to counterbalance the lost CMs. Cell therapy has conventionally been a modest procedure in which cells are directly injected into the myocardium [11,12,13,14,15,16,17,18,19,20]. These cells may function in various ways, including differentiation into cardiomyocytes, support for endogenous regeneration, and/or protection of the affected cells. Several cell types have been investigated for their regenerative ability, each with its own set of benefits and side effects [13]; among these are mesenchymal stem cells, bone marrow cells, and cardiac progenitor cells. These have been shown to improve heart function in preclinical studies and are currently being studied in clinical trials [21]. Cell therapy using mesenchymal stem cells and bone marrow cells has been effective to some extent, but these cells are unable to differentiate into cardiomyocytes due to their restricted differentiation potential.

The pluripotent properties of induced pluripotent stem cells do improve cardiac function and vascularization, but they still harbor the risk of teratoma formation [13]. Cardiomyocytes, smooth muscle cells, and endothelial cells can all be differentiated from cardiovascular lineage-specific progenitor cells which reside in the heart. Injection after MI demonstrated in vivo regenerative capacity, but progenitor cells were rejected by the host. Cell viability preservation and survival in the hostile environment of the infarcted heart, and further effective coupling to the existing myocardium, remain significant challenges. 

Another technique has recently emerged in which new cardiomyocytes are produced by inducing fibroblasts to transdifferentiate. Although the initial success rate of transdifferentiation was poor, it has the ability to introduce new cardiomyocytes even if the scar has completely developed [11]. Despite the fact that cell therapy has not yet proven to be as effective as anticipated, the findings obtained so far have provided further insights into how the damage in the infarcted heart can be handled. While mesenchymal stem cells and bone marrow cells do not contribute to the myocardium, they can still benefit the heart and its function via its secretive nature. Furthermore, cardiomyocyte progenitor cells positively impact the heart even though they do not differentiate into cardiomyocytes. As a result, other cell factors have a positive impact on the regenerative processes. Understanding regenerative processes and their underlying signaling mechanisms, as well as how cell therapy can affect these processes, is critical to reaping the benefits of cell therapy.

## 3. Cell Based Therapy 

In the tissue engineering and regeneration process, various types of cells are involved (Figure 3). Before using any cells, the key issues such as administration of immunosuppression and disease transmission to the host have to be addressed. Although autologous cell transplantation circumvents the use of immunosuppressants and holds a lower risk of disease transmission, the restricted supply hinders its application. Allogenic cells can also be used, but they require immunosuppression and pose a danger of disease transmission. Other disadvantages include the difficulty of collecting cells from donor sources, and of expanding their prior integration into the host. Furthermore, depending on the source of extraction (e.g., elderly persons or diabetic patients), autologous cells may have limited proliferation and differentiation [13,22,23]. Pluripotent stem cells such as embryonic stem cells (ESCs) and induced pluripotent stem cells (iPSCs) are cells that have the ability to self-renew and to give rise to any of the three primary germ cell layers, but not extra-embryonic tissues [24]. Studies employing stem cells in an animal model of cardiac injury and their outcomes have been summarized in Table 1.
Embryonic stem cells (ESCs)

ESCs are isolated from the inner mass cells of an embryo at the blastocyst stage. ESCs have the ability to proliferate for an infinite number of passages and can be differentiated into any cell type. ESC treatments with multiple induction cues can differentiate into cardiomyocytes or cardiac progenitor cells. Due to ethical concerns and the generation of teratoma, the use of ESC in clinical trials is ethically restricted. Some of the most challenging aspects of ESC research are teratogenic potential, obtaining pure lineage, and guiding differentiation to a particular lineage type [25,26,27]. To circumvent these constraints, genetic modifications, biological factor treatment, and diverse cultural approaches are applied. Chong et al. were the first to obtain a large number of cardiomyocytes from ESCs and use them to repair injured myocardium [24].
b.Induced pluripotent stem cells (iPSCs)

Takahashi and Yamanaka used viral vectors to generate induced pluripotent cells from somatic cells for use as personalized medicine. Since then, several studies have been published that have explained the ability of induced pluripotent stem cells (iPSCs) to differentiate into endoderm (e.g., hepatocytes), mesoderm (e.g., cardiomyocytes) and ectoderm (e.g., neurons), excepting extraembryonic tissues. Martens et al. and Yu et al. identified distinct cardiac phenotypes in infarcted mouse hearts using iPSCs [28,29]. These can replace the fundamental program settings through reprogramming with genetic factors and signaling molecules. As a result, a question about the experimental efficiency of iPSCs arises. Malignancies and oncogenes can also be induced in the host when employing viral vectors [23,30]. Maza et al. found that lowering the Mbd3 gene can enable all cells to acquire pluripotency, which is a key hurdle to employing iPSCs in clinical practice [31].
c.Adult stem cells (ASCs)

Autologous cells can be isolated from various sources (e.g., bone marrow, adipose tissue, etc.). Orlic et al. used transplanted bone marrow-derived cells (BMCs) to regenerate infarcted myocardium [32]. Several other investigations, however, cast doubt on this finding. Clinical experiments utilizing BMCs have shown short-term benefits as well as a higher survival rate [33]. When BMCs are coupled with growth factors, the benefits can last longer [34,35,36]. BMCs must be cultivated in vitro, but ASCs do not need to be grown. Cells are isolated from human fat tissue. The PRECISE, APOLLO, and RECATABI projects have conducted clinical trials [37]. A 3D polymer scaffold printed using a peptide gel and loaded with ASCs imparts mechanical strength to the already-dilated ventricle, according to the findings [38].
d.Cardiac stem cells (CSCs)

Cardiac stem cells can be isolated from a biopsy and then cultured in the lab. Smooth muscle cells, endothelial cells, and cardiomyocytes can be differentiated from undifferentiated CSCs [39,40,41,42,43]. Lineage tracing studies without a specific cardiac marker showed the existence of endogenous CSCs in the fetal heart; however, it was also pointed out in the study that there is a lack of data supporting the existence of CSCs in the adult heart. Moreover, recent studies revealed progenitors supporting regeneration of the damaged heart via secreting factors that rejuvenate the resident CSCs in order to counter balance the lost cells. However, massive damage requires a high number of cells to maintain homeostasis. Recent clinical studies conducted with CADUCEUS, using autologous cardiosphere-derived cells (CDCs), showed improved heart function [44,45,46].
cells-10-02538-t001_Table 1Table 1Description of the cells delivered to the heart by injection. This cell delivery approach has used various cell types including ESCs, iPSCs, MSCs, and CSCs.Initial Cell TypeTarget Cell TypeComposition of Delivery VehicleMode of DeliveryAnimal ModelsOutcomesLimitationsReferencesiPSCsCMsPolyethylene glycol hydrogelTrans-epicardialMI in nude ratsIncreased infarct thickness and improved muscle contentNo donor cell engraftment was observed[47]Mouse ESCs CMsPA-RGDS based gelTrans-epicardialMiceEngraftment and integration of mESC-CMs into host myocardiumimproved cardiac functionNo information available on cardiac remodelling [12]iPSCsCMsPBS solutionTrans-epicardialPost-infarcted swineEnhanced angiogenesis, reduced apoptosis, and blunted cardiac remodellingNo detailed information available on the engraftment of donor cell [48]MSCs****Self-assembling peptide hydrogels (3-D Matrix, Ltd.)Surface immobilization by spreadingLewis ratsAugmented microvascular formation and reduced interstitial fibrosisNo detailed information available on the engraftment of donor cell and CMs differentiation from MSC[49]MSCs****Si-HPMCTrans-epicardialLewis ratsShort-term recovery of ventricular function and attenuated mid-term remodellingNo detailed information available on the engraftment of donor cell and CMs differentiation from MSC[50]c-Kit overexpressing CSCs****PBS solutionIntracoronaryFischer 344 ratsPreserved LV function and structureIncreased cell dose was found to be harmful. Cell tracing or engraftment were not available in detail[50]CSCs****Matrigel and dimethylpolysiloxane mixture gelTrans-epicardialNOD-SCID miceImproved long-term retention of CSCs, cardiac structure and functionCell tracing or engraftment were not available [51]**** Studies were carried out to observe the improvement in cardiac function. The available report did not present specifics on the engraftment and in situ differentiation of delivered cells to mature CMs.
e.Skeletal myoblast cells (SMs)

SM cells can live in a low-oxygen environment better than other cells [52]. The ability to contract is the most crucial feature of these cells, as it allows them to contribute and adhere to beating cardiomyocytes [53]. These cells, however, are unable to integrate electromechanically with host cardiomyocytes due to a deficiency of connexin 43, a gap junctional protein. The results of the Myoblast Autologous Grafting in Ischemic Cardiomyopathy (MAGIC) clinical trials demonstrated that a pacemaker or a defibrillator with the incorporation of cells is needed for reducing arrhythmias. Phase II clinical trials of MAGIC involved the implantation of a cardioverter defibrillator along with SM via coronary artery bypass grafting [54]. Modifications based on the expression of connexin 43 are also being studied for avoiding myofiber arrhythmogenicity. Gap junction protein modification cannot tolerate arrhythmogenicity [55].
f.Umbilical cord blood cells (UCBC)

Umbilical Cord Blood Cells were isolated from the umbilical cord and have been extensively used for scientific research. UCBCs do not need ethical clearance, which allows for extreme flexibility when studying their biological dimensions. Even though these cells are less immunogenic, they show better reversing ventricular function ability in animal models [56]. Umbilical cord Wharton’s jelly-derived MSCs could also be considered a candidate cell type in cardiac tissue engineering, as they were shown to express cardiac-specific genes inherently without any manipulation [57]. Studies demonstrating the use of small molecule inhibitors to reinforce the enhanced differentiation potential of WJ-MSCs also highlighted the abundant supply of cells necessary for transplantation without any invasive procedures [58].
g.Amniotic fluid stem Cells (AFSCs)

AFSCs are prenatal stem cells having the potential to differentiate into cardiac cells or endothelial cells in vitro. These cells pose no risk of tumorigenicity or ethical concerns. In an immunosuppressed rat model, Yeh et al. demonstrated that these cells conserved ventricular wall thickness and improved heart functionality [59].
h.Cells Aggregates

Although stem cell transplantation is currently implemented clinically, it is difficult to accomplish minimally invasive injectable cell delivery while retaining high cell retention and animal survival. Strategies involving stem cell retention in the infarct region such as patch-based therapy and delivery of cell aggregates are being studied. Tang et al. demonstrated the safety and efficacy of encapsulating human cardiac stem cells (hCSCs) in thermosensitive poly (N-isopropylacrylamineco-acrylic acid) or P(NIPAM-AA) nanogel in mouse and pig models of MI. Unlike xenogeneic hCSCs injected in saline, injection of nanogel-encapsulated hCSCs did not elicit systemic inflammation or local T cell infiltration in immunocompetent mice. The developed thermosensitive nanogels can be used as a stem cell carrier: the porous and convoluted inner structure not only allows nutrient, oxygen, and secretion diffusion but also prevents the stem cells from being attacked by immune cells [60]. Compared to the traditional approaches of single cell injection, cell aggregate deliveries have demonstrated higher retention of cells and prevention of teratoma development [61]. Another such study on cell aggregates by Bauer et al. showed that the better survivability of these aggregates could be attributed to the imitation of the endogenous state by ensuring adequate cell-cell interaction [62]. A bioengineered 3D framework which enhances cellular contact while still allowing for certain cell ratios was developed by Monsanto et al. These injectable cardio clusters enhance adhesions and reduce cell loss [63].

## 4. Patch Based Cell Therapy Development

Various cell-based experiments have been conducted to treat MI. However, very few cells get migrated and incorporated in the infarct region owing to its hostile environment. Low oxygen supply to the infarct area prevents cells from being incorporated. A few pioneering studies have investigated the engineering of sheet-based cardiac patches constructed to harbor well-aligned and interconnected cardiomyocytes for successful implants to regenerate the myocardium [64]. Zimmermann et al. constructed myocardial tissue employing 3D models to mimic the native heart muscles, resulting in the restoration and improvement of cardiac function. Helfer & Bursac demonstrated a versatile framed hydrogel methodology to generate engineered cardiac tissue with enhanced mature functional properties. Therefore, it is well understood that in order to engineer a cardiac patch, the most important prerequisite is the cells that proliferate and gain functionality in the infarcted region. Different strategies involved in patch design for treating injured myocardium are shown in Figure 4 and described in the following section.

### 4.1. Properties for Patch Design

Scaffold materials are classified into synthetic or biologic and degradable or nondegradable conditional to their usage. Polymers used in the scaffold are subject to the composition, structure, and arrangement of their constituent macromolecules, which can be characterized into different types such as structural, chemical, and biological. Naturally occurring polymers, synthetic biodegradables, and synthetic non-biodegradable polymers are the leading types used as biomaterials. The successful fabrications of the 3D scaffolds utilized in MI can be broadly categorized into four groups based on their properties: chemical, electrical, mechanical, and biological.
Chemical: Surface properties (e.g., surface energy, chemistry, charge, surface area)Electrical: ConductivityPhysical: Mechanical competence (e.g., compressive and tensile strength), External geometry (e.g., macrostructure, microstructure, interconnectivity), porosity, and pore sizeBiological: Interface adherence, biocompatibility, biodegradation

The scaffold’s potential lies in its ability to serve as a cell surface receptor for cell differentiation, tissue formation, homeostasis, and regeneration, mimicking the natural ECM. Scaffold geometry plays a vital role in upholding highly interconnected porous fabrics of high surface density, thereby providing an increased surface-to-volume ratio, favoring cell attachment and proliferation [65].

#### 4.1.1. Chemical Properties 

Cellular adhesion and proliferation are surface-dependent properties, and the fabricated scaffold should facilitate the attachment of the cells. Altering the surface functionality by thin film deposition can aid in the better anchorage of cells wherein biomolecules viz. collagen, fibronectin, RGD peptides, and growth factors like bFGF, EGF, insulin, etc., are employed in scaffold design [66]. Microfabrication methods include manipulating topographic cues using lithography, which promotes cell organization into anisotropic 2D tissue layers [67]. The Angio-Chip scaffolds are formed with the help of POMaC incorporated nanopores. The microholes in the vessel walls help to increase permeability and permit intercellular crosstalk [68].

Scaffold degradation occurs as a result of physical, chemical, or biological processes. Enzymes involved in tissue remodeling also participate in the degradation of a scaffold. Consequently, scaffold dismantling and material dissolution occur through bulk or surface degradation. The polymeric biomaterials are degraded as a result of hydrolytic or enzymatic cleavage. The factors responsible for the degradation of polymeric biomaterials are the intrinsic properties of the polymer, chemical structure, the presence of hydrolytically unstable bonds, the level of hydrophilicity/hydrophobicity, morphology, glass transition temperature, the copolymer ratio, and the molecular weight of the polymer [65].

#### 4.1.2. Electrical Properties

The lack of electrical conductivity between the cardiac cells, i.e., synchronous beating between different parts of the patch, also needs to be addressed while designing biomaterials for cardiac tissue engineering. The nanostructures can be incorporated within the biomaterials to improve scaffold conductivity. This electrically designed bionic cardiac patch makes it possible to monitor and control engineered tissue functions after implantation [66]. Pairing tissue building blocks impregnated with specific DNA strands and complementary sequences with other building blocks governs their assembly to thicker tissues, while utilization of nanowired FETs allows for high sensitivity. Gold electrodes with a nanometric layer of titanium nitride have been used to increase the surface area, thereby improving cell adherence. The multifunctional electronic cardiac patch can monitor tissue activity in multiple locations. The electroactive polymers can release both positively and negatively charged molecules in response to the current where incorporated [67]. Incorporation of electrically conductive GNRs to the GelMA hydrogels has been designed to implicate the electrical conductivity of GelMA hydrogel constructs. GelMA–GNR hydrogel has an electrical impedance (1.35–0.36 kU) close to the physiological range [69]. It has been observed that conductive biomaterials, whether or not coupled with external electrical stimulation, enhanced the outcome of current tissue engineering strategies by improving cells or biomaterials’ native myocardium electromechanical integration [70].

#### 4.1.3. Physical Properties

Mimicking the mechanical properties of cardiac ECM is essential in order to provide biophysical cues to the cells. Acting as the natural cardiac microenvironment, the engineered patch produces the proper contractions. Thus, biomaterials with cardiomimetic mechanical properties such as PGS are used to promote the assembly of functional tissues with more native-like properties [66]. Rigidity and rheological parameters contribute to the mechanical stability of the scaffolds. It becomes crucial to retain the mechanical properties of the scaffold for its biological applications. These in turn help the regeneration of tissue. Parameters such as elastic modulus, flexural modulus, tensile strength, and maximum strain impart good mechanical strength to the scaffold and enhance its potential for several applications. The porous structure of the scaffold is entirely interconnected geometry. It is essential for cell ingrowth, uniform cell distribution, and enables the neovascularization of the construct. The crucial parameters when designing a scaffold are average pore size, pore size distribution, pore volume, pore interconnectivity, pore shape, pore throat size, and pore wall roughness. The pore size varies in different scaffold types, such as 5 μm for neovascularization, 5–15 μm for fibroblast ingrowth, 20 μm for the ingrowth of hepatocytes, 200–350 μm for osteoconduction, and 20–125 μm for regeneration of adult mammalian skin. It is essential to monitor the scaffold-pore interconnectivity to ensure oxygen mass transfer and nutrient transfer between cells [65]. PEG scaffolds have better elastomeric mechanical properties, with extended-release behavior and prolonged transgene expression [71]. GelMA hydrogel was engineered for surface micro-topographies that help to mimic the physiologically relevant myocardium function. Furthermore, it allows the cells to form uniformly dense and highly aligned cardiac tissues on GelMA–GNR hydrogels [72]. Collagen hydrogels, owing to their mechanical stiffness and high biocompatibility, are preferred candidates for cardiac tissue engineering. Further conjugating CNTs at subtoxic levels to collagen hydrogels demonstrate higher toughness, tensile stress, tensile strain, and electrical conductivity [73].

#### 4.1.4. Biological Properties

Tissue engineering is promising with biocompatible materials, and the scaffold or matrix should support cellular activity without hindering the signaling cascade. The chemistry and morphology of the materials used in the scaffold design as well as polymer synthesis and scaffold processing affect their biocompatibility. According to their biocompatibility various polymers are used in general medical applications, including PLA, PGA, PLGA, PDO, and PTMC [65]. Synthetic polymers possess advantages as their properties can be tailored to a specific application. They can be produced in large quantities, less cheaply compared to than biologic scaffolds though with enhanced shelf life. They demonstrate characteristic behavior in terms of tensile strength, elastic modulus, and degradation rate. PLA, PGA, and PLGA copolymers are widely used synthetic polymers in tissue engineering [65].

### 4.2. Biomaterials Used for Cardiac Tissue Engineering

Almost all normal cells in human tissue except for blood cells either reside in or adhere to the extracellular matrix (ECM). ECM provides structural support to the cells and contributes to the mechanical properties of the tissue. It also regulates cell behavior by influencing homeostasis, cell proliferation, cell shape, cell survival, differentiation, and migration. ECM also acts as a reservoir of growth factors and potentiates their actions while allowing remodeling during development and assisting differentiation [74,75,76,77,78,79,80,81]. Tissue engineering employs various cells, temporary scaffolds, and growth-promoting signals for achieving tissue regeneration and tissue repair [80,82,83,84,85]. Essentially, the scaffold should be an ECM analogue unique to the tissue of interest [80,82]. Although many types of ECM exist, all of them are primarily made up of a complicated assemblage of various polysaccharides and proteins that differ by tissue [86]. Several requirements should be met by cardiac tissue-specific construction. The constructs should mimic the native environment of heart muscles and should be viable during and after implantation. They should also help in the improvement of the systolic and diastolic functions of the injured myocardium [76,86,87,88]. The elastic and electrical properties of the scaffolds should be kept in consideration while designing the construct in order to achieve impulse conduction and contractile properties [86]. As a result, the ideal construct for cardiac regeneration should be mechanically robust and have high contractility and flexibility. It should also have angiogenic potential and induce vascularization after implantation. The constructs should also be non-immunogenic and electrophysiologically stable. There have been reports of different cardiac constructs based on various techniques. Biomaterial gels encapsulating cells, scaffolds encapsulating cells, cell films, and fibrous or porous sheets are among them [89]. The scaffold material chosen is a significant aspect that can influence the regeneration strategy’s success. The key polymer biomaterials used in cardiac tissue engineering are described in the sections below Table 2.

#### 4.2.1. Natural Polymers

The cell-biomaterial interaction is a crucial factor to consider when choosing a biomaterial for cellular delivery. Primary cells are adherent-dependent cells that die if the cell-matrix connection is disrupted. Many naturally occurring biomaterials exhibit the motif that is required for cellular contact and adhesion in this regard. Collagen, gelatin, decellularized tissue/organ, ECM, and silk fibroin are examples of such biomaterials [90,91]. These materials contain peptides which enable integrin mediated contact, leading to crosstalk between cell-matrix and cellular adherence via multimodal signaling pathways. Thus, in the case of natural biomaterials, cell-biomaterial interactions are more prevalent, which results in cell proliferation and differentiation. Since of intrinsic proteins, growth factors, and glycosaminoglycans (GAGs) are present, hydrogels obtained from decellularized ECM are a prominent natural biomaterial that demonstrates outstanding angiogenesis and cardiac regeneration abilities [92,93]. Hydrogels produced from decellularized hearts were used in a rodent model that exhibited better angiogenesis and myocardial function. Silk fibroin is a natural protein-based biomaterial generated from silkworms. Silk fibroin generated from non-mulberry silkworms contains an innate RGD peptide sequence that stimulates the adhesion of stem cells, proliferation, and differentiation [92].
(a)Fibrin

Fibrin is a natural polymer that FDA has approved for clinical use. It contains the amino acids arginine, glycine, and asparagine (RGD), which can help cells adhere together [94,95,96,97]. Fibrin can be made from fibrinogen monomers and thrombin polymerization, a proteolytic enzyme. Fibrin gels, on the other hand, have weak mechanical qualities and can shrink when injected into the heart. The injection of fibrin can cause intravascular thrombosis [98,99]. Using a blend of injectable fibrin with multiple types of cells (e.g., bone marrow cells, myoblasts, autologous endothelium cells, etc.) can outperform the efficacy of using just one type of cell [98,100].
(b)Chitosan

Chitosan is a carbohydrate that occurs naturally in chitin [101]. It is more accepted in the tissue engineering sector because of its biocompatibility and antifungal and antibacterial qualities. Because of its great temperature sensitivity, bioactive compounds are easily integrated into chitosan-based hydrogels [101,102,103,104]. Cardiomyocyte metabolic function can be improved by a thermoresponsive hybrid hydrogel made of chitosan, collagen, and QHREDGS (a peptide derived from angiopoietin 1) [105]. The hybrid hydrogel promotes cell survival, cell proliferation, and angiogenesis while enhancing myocardial wall thickness [105].
(c)Alginate

Alginate is a polysaccharide that can be made from both seaweed and bacteria. It is biocompatible to a high degree [86,106,107]. Its qualities can be tweaked by adjusting the concentration or molecular weight. Alginate was administered into the zone of infarction of the rat heart, resulting in a reduction in both scar thickness and systolic and diastolic cycle dysfunction. It also necessitates a purifying process prior to use in tissue engineering to ensure that alginate impurities do not create any adverse effects in humans. Clinical experiments with alginate hydrogels were reported by Anker et al., with patients having a higher mortality rate [108].
(d)Hyaluronic acid

Hyaluronic acid (HA) is a polysaccharide that is present in practically every cell in the body. It aids in the transport of nutrients into cells, maintains homeostasis, is nonimmunogenic, anti-inflammatory, and has numerous other beneficial effects on cell regeneration and repair [109]. The FDA approved it for research and certain human uses, and HA is commercially accessible in a cross-linkable form. Depending on its molecular weight, HA can initiate and contribute to a variety of biological processes. During degradation, low molecular weight HA that induces cell proliferation and angiogenesis is generated [110]. PEG-Sh4 biological development can also be functionalized [111]. Shen et al. evaluated various HA hydrogels with commercial chitosan, fibrin, and elastin hydrogels and observed that HA hydrogels were more biocompatible, less immunogenic and cytotoxic, and possessed better angiogenic properties than other hydrogels [112]. For cell attachment and proliferation, HA alone is inadequate. Cell adhesion can be achieved using hyaluronan hydrogels crosslinked with thiol-reactive poly(ethylene glycol) diacrylate [113,114].
(e)Collagen

Collagen is a vital component of a matured heart extracellular matrix (ECM), and it can help cardiomyocytes grow and survive in a natural way [115]. In animal models, commercially available collagen alone can improve heart function [116]. Collagen was administered using a catheter in a pig model, along with several types of cells, to demonstrate the possibility of a noninvasive delivery system using collagen [116]. Despite their advantages, collagen-derived gels are mechanically weak [117]. Using collagen matrix with induced carbon nanotubes (CNTs) helped to enhance rigidity, electrical, and mechanical abilities [118].
(f)ECM

The decellularized tissue scaffolds closely resemble natural ECM since they already possess its physiological and environmental characteristics [75]. Every tissue’s ECM has its own set of characteristics and components, such as proteins and proteoglycans. If the decellularized matrix is accessible, it is one of the finest choices for cardiac repair and regeneration [119]. The two most commonly available types of small intestine submucosa (SIS) derived injectable gels were investigated for heart repair in a mouse model. The gels had varied concentrations of fibroblast growth factors, and larger concentrations were more effective for heart regeneration [120]. Slow gelation and rapid breakdown are disadvantages of ECM-derived hydrogels [121,122]. Jefford et al. employed genipin for crosslinking porcine ECM hydrogels. Analysis showed that the degradation rate of genipin-crosslinked hydrogel was slower in vitro than gels without a crosslinker [123]. In an MI rat model, Efraim et al. investigated a functionalized chitosan decellularized porcine cardiac ECM using genipin as a crosslinker. The results demonstrated that this substance could dramatically improve heart function [124]. Gelatin, Matrigel, hair keratin, and laminin are examples of natural polymers. Natural polymers are biocompatible with host tissue and exhibit biological properties. On the other hand, natural polymers have some disadvantages, such as reproducibility concerns. Synthetic polymers are being generated to address the disadvantages of natural polymers while also improving the scaffold’s suitability for tissue engineering. ECM provides structural, mechanical, and biochemical signals to govern cellular processes, and the relationship between cells and their ECM has been widely investigated [125].

Scientists have been able to identify essential ECM components as well as the mechanisms by which ECM regulates normal cellular activities like migration and differentiation, as well as pathological events like cancer [119,121,126], fibrosis [127,128], and wound healing [78]. There is mounting evidence that changes in ECM mechanical properties significantly impact on cell structure and function. Excessive collagen-I deposition/crosslinking orchestrated by activated fibroblasts during fibrosis and tumor growth, in particular, is thought to have a role in aberrant mechano-sensing and atypical cell behaviors [129].

#### 4.2.2. Synthetic Polymers

Synthetic biomaterials have a number of benefits over their natural equivalents, including the ability to modify chemical, mechanical, and biological properties. Various synthetic biomaterials help in improving infarcted heart ventricular function when injected into the infarcted region [130]. When compared to sham, different types of poly(N-isopropylacrylamide) (PNIPAm) showed improvement in left ventricular end diastolic diameter (LVEDD) and decreased EF after injection [131]. Similarly, researchers found that injecting a poly (ethylene glycol) (PEG) hydrogel into saline-injected hearts reduced dilation by inhibiting an increase in LVED [132]. Synthetic biomaterials can be modified to fine-tune the target environment, resulting in improved bioavailability, cellular proliferation, and differentiation. A cell binding motif does not exist in pure synthetic biomaterials. Thus, it is necessary that cell binding motifs are introduced in these biomaterials covalently, which helps these synthetic biomaterials with cell adhesion. Similarly, covalent grafting can be used to introduce various functional groups on these polymers to link multiple pharmaceutical and biological molecules that would promote cellular proliferation and have differentiation and angiogenic properties [130,132].

Natural polymers have a wide range of compositions but are mechanically weak. As a result, a composite method has been applied to generate a composite biomaterial for cell transport made up of natural and synthetic polymers. Biocompatibility, mechanical characteristics, and the possibility of grafting molecules onto hydrogels generated from such composite materials are all improved [133]. An injection of fibrin and collagen combined with alginate, for example, hindered the migration of the infarct zone [134].
(a)Poly(ethylene glycol)

PEG is a frequently utilized synthetic polymer, attributed to its biocompatible nature [117]. PEG is a benign, nonimmunogenic polymer that may be customized by adding functional groups to its backbone [135]. It is soluble in water or organic solvents. It is nontoxic and non-immunogenic in nature. As a result, PEG is an appropriate polymer for heart regeneration. Since PEG is bioinert in nature, it does not mimic the microenvironment for cell survival. This constraint can be circumvented by crosslinking PEG hydrogels with natural polymers or employing bioactive compounds in the gels. PEG hydrogel containing cyclodextrin/MPEG–PCL–MPEG was developed by Wang et al. and supplied with erythropoietin (EPO) [136]. EPO is an antioxidant hormone that can protect the infarcted myocardium and minimize cell death. These developed gels demonstrated neovascularization when tested on rats, and resulted in a reduction in infarct size. PEG nanoparticles were also administered intravenously in the form of PEGylated liposomes (142 nm in size). This vehicle transports therapeutic compounds and releases them in a regulated manner [137]. For the binding of nanoparticles to the infarct region, overexpression of AT1 receptor (angiotensin II type1) was used, although the results were not very promising.
(b)Poly(glycolic acid) & Poly(lactic acid)

To tune the desirable qualities, polylactic acid (PLA) and polyglycolic acid (PGA) were combined with poly (lactic–co-glycolic acid) (PLGA). PLA and PGA are both FDA-approved and biocompatible materials. PLA is a non-cytotoxic suture material, and its degradation component, lactic acid, is also non-cytotoxic. Although PLA breakdown makes the microenvironment slightly acidic [138], PGA is a non-cytotoxic thermoplastic. Neither PLA nor PGA, on the other hand, can equal the flexibility of heart tissue. As a result, they are mixed with other polyesters. Using a fibrous membrane made of electrospun PLGA, cardiomyocytes could be oriented to the direction of nanofibers [139]. The delivery of porous PLGA beads seeded with hAFSCs to a rat infarct model using a vehicle or “Cellularized Micro scaffolds” resulted in good cell retention [140]. To increase biological characteristics, PLGA could be mixed with natural polymer laminin or with carbon nanofibers (CNF) to induce conductivity [141,142].
(c)Poly(ε-caprolactone)

At body temperature, poly(ε-caprolactone) has a low glass transition temperature and behaves like rubber or elastic [143]. A 3D structure made up of up to five layers of electrospun PCL nanofibrous mats were evaluated for new-born cardiomyocyte culture, and the layers were able to form electrical connections and beat in time [144]. It is usually mixed with PLA or PGA copolymers. A biodegradable porous scaffold made of poly-glycolide-co-caprolactone (PGCL) was employed in a rat infarcted myocardium model to distribute bone marrow-derived mononuclear cells (BMMNC). BNMC was transported from the scaffold to the implant, and neovascularization was seen [145].
(d)N-isopropylacrylamide (poly(N-isopropylacrylamide)) (PNIPAAm)

Thermosensitive polymer PNIPAAm at 32 °C possesses a reversible transition point. It is suited for biomedical applications because of its solution-to-gelation (sol-to-gel) transition point [146,147]. Hydrogels based on PNIPAAm can support cell co-cultures, which are believed to aid cardiac tissue regeneration [148]. Navaei et al. created a hydrogel containing 3D PNIPAAm-gelatin and co-cultured neonatal rat ventricular myocytes (NRVMs) and cardiac fibroblasts (CFs) [149]. As a result, they discovered that co-culturing improved cell contact and homogenous beating when compared to monoculture. Although PNIPAAm has a number of benefits for cardiac tissue engineering, its biodegradability is a concern [150]. Scientists have devised a number of solutions to this problem. With poly (NIPAAmco-2-hydroxyethyl methacrylate (HEMA), Fan et al. created an acrylate oligo-lactide (AOLA) degradable hydrogel [151]. They discovered that adding HEMA into poly (NIPAAm) caused the hydrogel to break down into a byproduct that is water soluble at body temperature. The most intriguing aspect of PNIPAAm is that it can be conjugated with carbon nanotubes (CNTs) to make the scaffold conduct [152,153]. In a rat MI model, Li et al. employed PNIPAAm in an injectable form in combination with single-walled carbon nanotubes (SWCNTs) to produce brown adipose-derived stem cells (BASCs) [154]. As a result, there was a noticeable increase in cell integration. Materials made of aniline, due to its electroactive and antioxidant capabilities, are a desirable material for cardiac tissue engineering [155,156]. A chitosan graft-aniline tetramer (CS-AT) and poly (ethylene glycol) (PEG-DA) hydrogel were developed by Dong et al. [157]. Electrical cues were transmitted due to the presence of polyaniline in the polymer backbone. Murine myoblasts and adipose-derived MSCs (ADMSCs) displayed vitality and proliferation in vitro after being encased in the hydrogel.
(e)Hybrid gelatin methacryloyl (GelMA)

The superior biocompatibility and controlled biodegradability of hybrid gelatin methacryloyl (GelMA) hydrogels makes them ideal for tissue engineering [158,159]. GelMA is made by combining gelatin and methacrylic anhydride. Hydrogel strength and stiffness can be modified by altering the amount of methacrylic anhydride [158]. GelMA/PEGDMA (PEG di-methacrylate) was used to encapsulate C2C12 myoblasts with stiffness ranging from 12 to 42 kPa by Li et al. [160]. This combination has the potential to cause muscular myofiber development. This crosslinking was created by exposure to UV radiation, and it was then employed to build blood vessels [160]. UV light, on the other hand, can be harmful to the heart [161]. To circumvent this constraint, Noshadi et al. created a cross-linkable GelMA hydrogel that can be exposed to visible light. Neonatal rat ventricular myocytes (NRVMs) were cultivated on top of this hydrogel for at least seven days, and the cells maintained their cardiac phenotype [161].

### 4.3. Delivery Strategies of Cells from Patch

VEGF-encapsulated MSCs are used to treat MI tissue, which helps in the improvement of cardiac function by angiogenesis based on the tropism of the MSCs to the MI area [162]. When encumbered with stem cells, hdECM is used as bio-ink in the 3D printing of pre-vascularized and functional multimaterial structures. The printed structure is comprised of the spatial patterning of dual stem cells, which are associated with enhanced cardiac function, decreased cardiac hypertrophy and fibrosis, elevated migration from the patch to the infarct area, neo-muscle, and capillary formation with the improvement in cardiac function. The hdECM potentiates in epicardial-mediated cardiac tissue. It is regenerated following the migration of WT1 positive progenitor cells using the EMT process [163].

Polymeric scaffold-mediated viral delivery showed that the release period could be extended from several days to one month if the molecular weights and concentration of the PEG varied. Sustained viral delivery from the core sheath fibers is facilitated by the solid fiber sheath that uses PEG as a porogen [71].
(a)Invasive method

The cardiac patch developed requires open-heart surgery to implant the patch, either by suturing [164,165] or by applying bioglue to the patch [166]. A PCL/gelatin patch incorporated with MSCs activated endogenous cardiac repair by enhancing the survival of MSCs and their HIF-1a, Tb4, VEGF, and SDF-1 expression and decreased CXCL14 expression in hypoxic and serum-deprived conditions. The engrafted MSCs were examined for survival and distribution, and it was observed that the engrafted MSCs migrated across the epicardium and into the myocardium. The epicardium was activated, and EDCs migrated into the deep tissue, which was differentiated into ECs and SMCs, with a few differentiated into CMs [167].

Early vascularization in the ischemic heart is critical for a better outcome. The delivered iVPCs were grown on polymer microbundle scaffolds made up of PLGA, showing beneficial effects on cardiac repair and recovery. When the iVPCs were integrated into a micro-bundle scaffold of PLGA that served as a carrier, the treatment efficacy improved [168].
(b)Minimum Invasive Method

The application of the cardiac patch to the heart requires open-chest surgery, which is traumatic. To overcome this issue, different groups came up with different ideas. Tang et al. designed a biospray using platelet-fibrin gel “paint” which polymerizes in situ with a minimally invasive procedure. They demonstrated that spray treatment improved cardiac repair and weakened cardiac dysfunction after MI [169]. Miles et al. engineered an elastic and microfabricated scaffold that could be delivered via injection. The scaffold material could recover its initial shape while maintaining the viability and function of the cells, as it has occurred after being delivered through an orifice as small as 1 mm. They demonstrated significant improvement in cardiac function in an MI rat model [170]. Brisa et al. designed an injectable RTG and functionalized it with CNTs. It transitions from a solution at room temperature to a 3D gel-based matrix shortly after reaching body temperature, which supports long-term CM survival, promotes CM alignment and proliferation, and improves CM function compared to control. Functionalized RTG with CNT renders both topographical and electrophysiological cues for native CMs. It promotes long-term survival with a more aligned cell organization, and suppresses fibroblast proliferation [171]. VEGF-encapsulated MSCs are prepared by self-assembling gelatin and alginate polyelectrolytes. It is a minimally invasive therapy to treat MI [162].

### 4.4. Advantages and Disadvantages of Patch 

#### 4.4.1. Advantages

The cardiac patch designed from natural or synthetic polymers provides a niche to stem cells [66,67]. The cardiac patch acts as a reservoir for growth factors and viruses for gene therapy [71]. The electroconductive cardiac patch helps in the proper alignment of cardiac cells while aiding in vitro monitoring [68]. The cellular component incorporated in the patch system is anticipated to ameliorate the pre-existing inflammatory milieu and subside the at the tissue damage site. Particularly, MSCs are reported to exist in the various tissue sources, having distinct proliferation and differentiation potential. Earlier studies were reported about the repopulation of worn-out populations at the tissue damage site; however, more recent studies have shown that the paracrine secretory profiles of MSCs are responsible for their beneficial effects, such as anti-inflammatory and immunomodulatory features. Therefore, the environmental sensing potential and its further responsive actions via the secretion of membrane-bound vesicles known as exosomes (which harbor growth factors, cytokines, and miRNAs), have contributed to establishing physiological homeostasis at the site of tissue damage [172]. 

#### 4.4.2. Disadvantages

Under significant stimulatory conditions, the secretory profile of MSC-derived exosomes have not been completely streamlined for immediate harnessing from bench to bedside. Systemic targeted delivery of MSC-derived exosomes for the amelioration of hypoxia, apoptosis, and inflammatory milieu for accelerated myocardial regeneration may warrant its clinical utility. Therefore, we recommend in-depth research on the patch-based cell and cell-free approaches using humanized models and clinical trials, and highlighting its importance in improving the quality, safety and efficiency of future cardiac therapy [172]. Despite successes in constructing different types of tissue-engineered cardiac patches, clinical use has yet to be achieved due to a lack of in vivo verification of feasibility.

## 5. Microfluidics Based MI Research

Microfluidics deals with the handling of very small volumes of liquid, typically in a length scale of a few micrometers. Since its development in the early 1980s, it has been discovered to be an effective tool with a wide range of applications, from inkjet printers to LOC systems. Over the last two decades, microfluidics has been used in many biological fields such as genomics and proteomics, point-of-care diagnostic systems, biohazard detection, and more. Its recent developments have enabled tissue engineering research to be done in a more cost-effective and highly sensitive way. Research related to MI has been explored using microfluidic devices as it becomes feasible to create a 3D tissue structure and study it in a dynamic condition mimicking both healthy and pathophysiological states of the heart [173].

Microfluidics emerged from the conventional silicon and glass micro-machining process. With the advent of photo-lithography and other BioMEMS fabrication techniques, the whole process has become more user-friendly. The range of materials used for fabrication is vast, and includes different polymers, silicon, silicon-based materials, metals, etc. Materials are selected based on properties like rigidity or flexibility, optical transparency, biocompatibility, and reactivity to reagents. PDMS is a common material used for microfluidics because of properties like high oxygen permeability, ease of fabrication, and biocompatibility. As cell behavior also depends upon the topography of its environment, the surface chemistry of these materials plays a vital role in cell culture. The biocompatibility of these surfaces can also be increased via processes such as plasma deposition [174].

Cell culture work close to the physiologic environment is possible in a microfluidic chip, with better control over the process parameters. Such work has led to the use of human samples for research and reduced dependence on animal models for drug discovery and therapeutics. The small sample size requirement for microfluidic devices helps in handling costly samples and reagents. Miniaturized 3D tissue models can produce better studies on physiological systems and their behavior for toxicity assay studies. The flow conditions in a microfluidic device are very much essential in cellular studies, as the physiological environment is seldom static. 

The microfluidic LOC device has also been designed to easily generate different physiological, mechanical, or electrical forces on the culture, which are complicated in macro bioreactors. The whole system can be designed and optimized through different available computational software (e.g., Comsol, Ansys, etc.). These devices can also be linked to external analytical devices in order to study real-time cell behavior (Figure 5) [175]. 

Various peculiar properties come into existence when the fluid flows in a microfluidic network, which are generally suppressed in macrosystems. The fluid flow is always in the laminar regime, as the Reynolds number stays very low. Essentially, this prevents the mixing of two adjacent layers of fluid. Diffusion becomes a major cause for mass transfer, which is used for easily generating a stable spatiotemporal chemical gradient on LOC devices; as the surface-to-volume ratio is very high, surface forces such as capillary forces become dominant. These properties come into play in the efficient designing and control of microfluidic devices.

### 5.1. Microfluidics for Cardiac Cell Biology

Among all CVDs, MI is the primary cause of mortality in the world. MI is caused due to occlusion in the coronary artery, leading to reduced or no blood flow to cardiac tissues. It causes reduced functionality of CMs, interstitial fibrosis, and ultimately cell death. Current therapeutic approaches to treat MI include the use of drugs to alleviate the symptoms or the use of stents to increase blood flow that can extend the heart’s life. However, these approaches do not consider the repair of tissue or compensate for the loss of cardiac cells [46]. Recently developed cell-based and patch-based methods try to fill this gap, and concentrate more on developing techniques to repair and regenerate the infarcted area with a healthy network of blood vessels. To design a patch efficiently, there is a need to closely understand cardiac physiology. The heart is a very dynamic system, with different physical and chemical cues that control its functions. At this stage the role of the microfluidic system becomes important, as this novel tool can be efficiently utilized in MI diagnosis and the study of differentiation of stem cells to cardiac cells. Furthermore, it helps to generate close to physiological conditions in cell culture models, which can produce highly applicable results in tissue engineering research. Co-culture of different cardiac cells, which is necessary in order to understand cell-to-cell interaction, becomes easy with microfluidic devices. The use of human cells in these kinds of micro biomimetic environments makes them replicate the myocardium of an individual human patient. While such a myocardium-on-a-chip is currently intended as a tool for in vitro analyses, the platform may provide the foundation for future development of superior tissue engineering constructs for cardiac tissue replacement.

### 5.2. Application of Microfluidics in MI Research

Despite the scarce publications relating to the study of MI in microfluidics, some significant research is being done in microfluidic devices to explore the basic as well as the applied aspects. Recently, it was revealed by a single-cell microfluidic study that human iPSC-derived CMs secreted cytokines that ameliorate conditions of MI [180]. Such results were further confirmed by an in vivo study, where the role of cytokine secretion was established in a mouse model of acute MI. This opens up the opportunity to explore and study the role and effect of various cell types which are potential candidates for differentiating into mature CMs and/or have a therapeutic paracrine effect. Another use of microfluidics related to MI research was discovered when the role of inflammatory monocytes was studied in MI patients. 

Phonotypic study of monocytes revealed that CD11c/CD18 was an inducible integrin whose expression correlated with a monocyte inflammatory state in MI patients. It was observed that under shear conditions simulating the low dynamic of blood, CD14^++^ CD16^+^ monocyte adhesion was double that of the healthy human subject. It is essential to recruit monocytes or inflammatory cells in ischemic tissue, which is done with the WBC components from circulation. Such cells are attracted when they sense chemicals secreted in response to injury, tissue necrosis, significant cell death, or another inflammatory cellular/tissue response. This dynamic could be easily simulated by a microfluid-based LOC platform [181]. Cellular analysis in a microfluidic platform is more similar to the dynamic conditions of the physiological state. Cell encapsulation in hydrogels enhances their survival in transplanted tissue and makes regeneration easier, and also aids in the production of prohealing cytokines and extracellular vesicles. Microfluidic cocooning has several advantages over vortex cocooning, such as limited consumption of biomaterials, precise cocoon size and cell number, extremely limited shear stresses on cells during cocooning, high individual unit throughput, etc. [182]. Another key benefit of the microfluidic system is the ability to pattern cell and extracellular matrix (ECM) at the cell length scale. Patient-derived cells can also be used, as they require fewer numbers of cells than conventional methods. Both 2D and 3D structures can be effectively generated in the microfluidic chip with the help of surface coating or the use of hydrogels [183]. Such devices can be made to mimic cardiovascular physiology (including ECM structure, cell composition, electrophysiology, heart mechanics, vascularization, etc.) for drug discovery and interventional strategies. Ischemia in the myocardium develops due to thrombus formation inside the coronary artery. Sono-thrombolysis is an effective method for the dissolution of a thrombus in blood vessels. Flow focusing techniques in microfluidics can form microbubbles that can be used for mechanically disrupting the thrombus. Such a technique has been used in a rat model to treat cerebral infarct with Nitrogen microbubbles, and shows a 50% reduction in infarct after thrombus dissolution [184].

Following MI, there is an irreversible loss of CMs that results in the alteration of electrical propagation in the heart. Stem cells are seeded onto the heart directly or through a patch to compensate for the loss of CMs, which helps repair the infarcted area. Mesenchymal stem cells are one of the important cell sources for cardiac tissue engineering. It has been found that their harvesting efficiency is higher when they are cultured in dissolvable gelatin-based microcarriers [185]. The surrounding microenvironment of cardiomyocytes needs to be controlled for appropriate growth and regeneration of the ischemic tissue. Different factors like cell seeding density, type of hydrogel, the percentage and type of non-myocytes, and electrical conditioning can be studied in detail on a microfluidic platform [186]. 

Different research groups have developed many animal models to test the efficacy of cell types and drugs in myocardial regeneration. However, the results from these animal models cannot be replicated in a patient heart. The development of a cardiac muscle model in microfluidics eliminates all these obstacles. These chips can incorporate on-chip analyses of the electrical conductivity of different cell types. The Laser patterning technique comes in handy as it can arrange cells in desired directions to increase the contact between the stem cells and CMs. Such chips can replicate the in vivo conditions very well. Along with this, it is also possible to test the importance of cell alignment and cell-to-cell contact in stem cell delivery on cardiac tissues [187,188,189]. The applications of microfluidic techniques for CVD research are shown in Table 3.

### 5.3. Microfluidic 3D Culture Models

Cells in our body reside in a complex 3D environment made out of cells and the ECM. Crucial processes such as nutrient transportation and cell-to-cell interaction depend upon the spatial arrangement of cells and the properties of the ECM. The spatiotemporal gradient of nutrient and oxygen concentration is important in tissue maintenance and regulation. Moreover, the heart is such a complex organ, having multiple cell types and specific cell arrangements, that it becomes challenging to recapitulate the in vivo physiology of the heart in 2D cultures. To obtain culture conditions close to physiological ones, the generation of a 3D environment becomes essential. In conventional culture conditions there are few models which can capture the physiology of infarcted heart tissue using cardiac organoids that mimic the oxygen gradient and adrenergic stimulus in MI [190]. Microfluidics give a vast range of options in which a micro 3D environment can be created. Hydrogel-based micro 3D tissues can be formed inside a microfluidic chip. The 3D model can have a perfusion-based distribution of media or can have a diffusion-based media flow that mimics the endothelial barrier in cardiac tissue [46,191]. Cardiac bodies are other options for microfluidic cell culture models, which do not require a hydrogel mixture [192]. Due to in vivo-like environment generation 3D culture promotes more interactions between cells, allowing them to remain in optimal conditions for proliferation. Drug cytotoxicity is also significantly lower for 3D than for 2D culture.

The efficiency of reprogramming human somatic cells to iPSCs can be improved up to 50-fold in a microfluidic environment [193]. It also consumes fewer reagents than a six-well plate culture. Patient-specific hiPSCs can be differentiated without cell expansion to produce functional CMs in as little as a month’s time using such devices. Differentiation and proliferation from cells like MSCs, ESCs, or iPSCs to CMs can be achieved very economically and with better efficiency. 

Cellular scaffold generation is an important part of developing a patch for MI or studying cell behavior in 3D constructs. The strength of the scaffold, along with its porosity and biodegradability, are essential elements in its design. Micro-bioprinting techniques help in generating the desired scaffold preparation using different polymeric materials and extracellular proteins. Precise loading of scaffolds becomes easy with this technique allowing the formation of highly viable and functional in vitro constructs with excellent resolution. This technique also allows the use of heterogeneous bioink [194]. Easy fabrication methods make this scaffold matrix highly tunable in forming a network of blood vessels [68]. Researchers have also combined microfluidic technology with 3D micropatterning to produce unique cell-laden structures, which can acquire contractile stresses generated by cardiac cells [195]. Biopolymers have also been used as scaffolds that have similar biochemical components as native tissues [196].

### 5.4. Implementation of Physical Forces

Cells in the myocardium are always subject to physical forces like hydraulic pressure, mechanical and fluid stresses, electrical field forces, etc. To develop a treatment method for the CVDs, the effect of these forces on cardiac cells and stem cells used for treatment needs to be studied extensively. A mechanical cue like cyclic stretch is a fundamental aspect of heart tissue that can vary from normal to pathological. Stretch at 1 Hz and 10% strain has been shown to yield significantly less cardiomyogenic differentiation of ESC culture [197]. This may be caused by disruption of the cardiomyogenic differentiation process prior to the expression of an MHC, a late-stage marker for cardiogenesis. It can be concluded that the strain magnitude, frequency, direction of strain, duration of stretch application, and the stage of differentiation at which the stretch is applied are all variables that are needed to be systematically investigated. Except for cyclic stretch, other mechanical factors such as hydrostatic pressure and fluid shear stress also play an important role in cardiac physiology. An important thing to note is that these variables change drastically in pathological conditions. Recently, a device to recreate the mechanical loading conditions of the left ventricle (LV) was developed [198]. This device could produce different pressure conditions that simulate normal and pathological conditions like hypertension, hypotension, tachycardia, bradycardia etc. After occlusion of the coronary artery, ischemia develops in the myocardium. While it is necessary to restore blood flow to the ischemic tissue, rapid perfusion can also cause tissue damage called ischemic reperfusion injury (IRI). It is difficult to study IRI in human models, although there are in vitro models which could recapitulate the oxygen concentration and pH condition in IRI [199]. Such models also show that cardioprotective therapeutics and ischemic preconditioning have the ability to reduce IRI [200]. Microfluidic models, with their greater flexibility and control, can model the in vivo environment and reperfusion issues in a better way. Results from such a model show that reperfusion indeed activates endothelial cells with a higher expression of I-CAM 1. Reperfusion can also lead to endothelial injury [201].

Embryonic cardiomyoblast line (H9c2 cells) studied on microfluidic devices showed organized F actin alignment similar to in vivo conditions with cyclic strain. This device could also be used for the co-culture of different cells of cardiac tissue that helps to study their physiology and cell-to-cell interactions. Uniaxial cyclic strain could also induce better electrical and mechanical coupling between cells, with a remarkable increase in junction complexes [202]. It is necessary to understand the behavior of all types of cells present in the heart in order to develop fully functional cardiac tissue. Cardiac fibroblasts (CF) are one type of cell which are primarily involved in cardiac tissue remodeling. CFs have shown increased elongation with stretching [203]. Their proliferation behavior can also be peculiar at different strain intensities. 2% stretching has shown higher proliferation, whereas 8% stretching gives rise to time-dependent proliferative behavior with a drop in the number of mitotic cells after 72 h. Studies like this can give insight into the pathological evolution of fibrotic cardiac disease. The extracellular environment can also direct cell behavior; this is true for both cardiac cells and the pluripotent stem cells from which they are differentiated. Microfluidics makes an excellent tool for such studies. It is very easy to pattern the cell attachment surfaces through micropatterning technologies [204]. Microfluidics offers numerous substrate choices, as mentioned earlier, which makes the study of substrate stiffness on cell physiology convenient. The cellular environment can also be changed by using different hydrogels and ECM proteins in micro cardiac tissues.

Through microfluidic techniques, the contractile efficiency of micro heart tissue can be measured by the deformation of a micropillar array on a chip [205]. Different alignments of cardiomyocytes can be tested in order to understand the contractile efficiency with cell orientation [206]. Electrical stimulation is another cue that stimulates CM function and the differentiation of pluripotent stem cells into CMs, and also helps in the maturation of iPSC-derived CMs. Micro-patterning of nontoxic electrodes like ITO in microfluidic devices offers an excellent platform for such studies. These electrodes have excellent electrical conductivity along with optical transparency. Therefore, they can help in designing a highly efficient system that can conveniently provide on-chip imaging [207]. In another work, a microelectrode array platform was fabricated for recording the beating rate and conductive velocity of cardiomyocytes. In this device, the cardiac monolayer beat in a synchronized fashion, and the conduction velocity was close to the physiological value [208]. Different methods can measure the beating of heart tissue. A cell culture substrate was generated in a novel microfluidic device based on the adhesive properties of polyethylene glycol diacrylate (PEGDA) and gelatin acryloyl (GelMA). Owing to this adhesive property, the cardiomyocytes adhered only to a specific region. During beating, the stretching of PEGDA substrate triggered a shift in its color, which can be visualized through optics. Reduced graphene oxide (rGO) was doped on this substrate to increase the contrast of light [209].

**Table 3 cells-10-02538-t003:** Use of microfluidics platforms in the study of various aspects of CVD, and applications in cardiac tissue engineering.

Device Function	Cell Source	Techniques Used	Chemical or Physical Cues Studied	Scaffold Used	Fabrication Technique	Important Observations	References
Differentiation to CMs	ESCs	External motor for stretching the microfluidic device	Uniaxial cyclic mechanical stretch	2D culture	Lithography	Reduction in cardiogenesis	[197]
	hESCs	Micropatterned surface generation through direct micro contact printing	------	Micropatterned fibronectin hydrogel	------	Display of beating foci earlier than non-patterned substrates	[204]
Drug toxicity testing	Human iPSC-CBs	Micro niches to trap CBs in microchannel, Perfusion based system	Veparamil, Quinidine, Doxorubicin	No external scaffold	Standard photo lithography	3D environment showed different effect on beating frequency of cells	[193]
	Human CMs	Micropillar based system to prohibit direct contact between 3D cell matrix from media flow, diffusion-based transport	Isoproterenol	Puramatrix hydrogel	PMMA micromilling	Cell viability appeared better in 3D culture	[192]
Contractile stress measurement	Neonatal rat ventricular myocytes and human iPSC derived CMs	Electronic quantification of stress through Cantilever deflection measurement	Isoproterenol	3D printed matrix of PDMS with polyamide electrical network	Multimaterial 3D printing	Positive chronotropic response to drug similar to engineered NRVM microtissues and ESC-derived CM tissue	[196]
	Neonatal mouse CMs	Stress measurement by use of PIV technique to capture nanoparticle displacement coupled with finite element method.	Epinephrine	Sandwich of GelMA hydrogel and polyacrylamide hydrogels	3D patterning	Increased frequency and amplitude of contraction cycles	[195]
Generation of in vitro constructs for tissue engineering application	Neonatal rat CMs	Coaxial needle extrusion system	------	GelMA	3D printing	Generated complex heterogenous structures with single bioink extruder	[194]
Hydraulic pressure and mechanical strain condition generation	H9c2 cells	Use of peristaltic pump coupled with pneumatically actuated valve to generate pathological heart conditions	------	------	PDMS molding	Organized F actin alignment similar to in vivo	[198]
	Neonatal rat CM	Pneumatic deflection of thin PDMS membrane to generate stretch	Uniaxial cyclic stretch	Cell laden gel	Lithography	Superior cardiac differentiation with better electrical and mechanical coupling	[202]
Effect of electrical field on proliferation and differentiation	Neonatal rat CM	------	Square monophasic electrical pulses	2D cell culture on collagen coated matrix	Laser ablation of ITO coated glass slides to generate electrodes	Cell aligned in the direction perpendicular to the electric field	[207]
3D environment mimicking shear protection from endothelial barrier	hiPSC derived CMs	------	Verapamil, Isoproterenol, Metoprolol, E-4031	------	Two step photolithography process	IC50 and EC50 values were more consistent with the data on tissue-scale references	[191]

### 5.5. Drug Discovery and Disease Modelling

Diseases related to cardiovascular tissues are the most prevalent diseases worldwide. The major goals of tissue engineering are to develop, screen, and test drugs that can either cure these ailments or at least maintain healthy cardiovascular function. Conventionally, a 2D culture system is used to study the effects of drugs on cardiac cells, but this system fails more often than not owing to the reason that 3D in vivo physiological conditions change cell behavior. Because of this, many of drugs show negative results and are rejected in preclinical models. Microfluidic devices are the most useful tool in modern-day drug discovery research for generating 3D cell culture conditions and easily introducing in vivo dynamics. These microfluidic models can be used for drug discovery and for toxicity testing of different drugs in cardiac tissues. These microtissue models can be generated by different techniques. In recent work, such a model of cardiac bodies (CBs) was developed from induced pluripotent stem cell (iPSC) clusters [192]. Injection of hydrogels inside the infarct region has been used for myocardial regeneration, but hydrogels lack a porous structure that can support the regenerating tissue. Through microfluidic manufacturing, microporous particles can be generated with microgel building blocks, which will contain drugs to induce tissue generation and support tissue growth with their porous structure [210]. Numerous micro niches can be formed inside the device that can accommodate numerous tissue samples for drug testing studies, reducing the number of experiments compared to the macro system. Researchers have studied the effects of different drugs such as verapamil, quinidine, and doxorubicin, epinephrine, etc., on the function of cardiac cells. Microdevices can be useful in producing a 3D network of cardiac cells with proper in vivo like alignment and cell-to-cell interactions. By mimicking the cardiac microenvironment closely, a microphysiological system can also be designed to establish in vivo conditions like cell alignment in the 3D microtissue, diffusion-mediated nutrient transportation, and micro-circulatory network. Mathur et al. could generate endothelial barrier-like physiology inside their device by patterning micropillar-like structures in it [191]. The introduction of all these features has shown significantly improved drug response results compared to conventional 2D cultures. Treatment with pharmacological agents like isoproterenol and clinical drugs like verapamil showed comparable beating behavior of these cardiac tissues to that seen in acute experiments performed on human ventricular heart tissue slices.

Production of drug encapsulated microspheres can be formed using microfluidics, with controllable size distribution. Proteins like ACE2 with a short half-life and poor stability can be encapsulated in alginate microspheres with microfluidic electrospray techniques [211]. Droplet formation techniques in microfluidics can be used to form microcapsules with a drug in the inner core and a polymeric shell to control its release. Drugs with a higher concentration can be loaded in the core for sustained delivery. In the acute myocardial infarction model of a rat, drugs like VEGF and PDGF have been delivered through this method. It was found that the repair of the infraction was significant with respect to the reduction in ventricular wall thickness and fibrosis, and the percentage of scar tissue was reduced to 11.2%, compared to 21.4% with saline [212].

Another aspect of drug discovery is cytotoxicity studies on other tissues in the body apart from for the primarily targeted tissue. After the metabolism of the drug molecules, byproducts are released which can affect the functioning of other tissues. Microfluidics allows the formation of a close circulation loop between different tissues on the same chip that can predict the cytotoxic effect of drugs and byproducts, along with their efficacy. Researchers have used cells from tissues such as the heart, liver, and kidney on the same chip to predict the fate of the drug in the human body and its metabolism and excretion mechanism. The cardiotoxicity of other drugs such as doxorubicin can be studied on a heart and liver organ on a chip [213]. The investigation of cytotoxicity and efficacy can be precisely predicted on such platforms to achieve personalized medicine and patient-specific care.

The bioelectrical activity of the heart is a measure of healthy cardiac function. During myocardial infarction due to hypoxia and nutrient deprivation, the electrical physiology varies. In a recently developed heart-on-a-chip system, both extracellular and intracellular potential have been measured in hypoxic conditions with the help of top-down lithography and nanofabricated platinum electrodes, respectively [214]. It was found that initial hypoxia caused tachycardia which then transformed to beat rate reduction and eventually arrhythmia. Intracellular bioelectronics showed narrowed action potential after the introduction of hypoxia.

### 5.6. Point of Care Devices and Disease Diagnosis

Disease diagnosis is an important stage in cardiac tissue engineering, in order to find out the type and nature of the ailment. Microfluidics allows a simple and fast method to detect diseases. The devices can be more cost-effective than conventional systems, and they have the added benefit of less sample volume usage. The use of nanoparticles as a signal label for the detection of biomarkers such as NT-proBNP and cTnI in microfluidic devices is highly suitable for immunoassays [215]. Although the commercial availability of such devices is very low (e.g., Alere Triage R system, Abbott i-STAT R system), the technology can be optimized considering the type (magnetic or nonmagnetic) and size of nanoparticles, as well as methods to reduce noise due to nonspecific adsorption, etc.

CVDs are the main contributor to global deaths per year. This cause surpasses the sum of deaths due to cancer and infectious diseases. Most of the deaths occur in low-income countries where disease diagnosis and treatment are costlier. Microfluidic technology can bridge this gap and help everybody get proper medical attention. Microfluidic chip-based immunoassays are more sensitive and can give better results. Currently available centrifugal microfluidic devices on the market, like Abaxis Piccolo Xpress, can use capillary and centrifugal action together and can separate plasma and perform fluid mixing and spectrometric measurement right on the microfluidic compact disk [216]. Biomarker detection through chemiluminescent, electrochemical, and optical methods is currently in the research stage and can enrich microfluidic technology for cardiac disease detection. Paper-based microfluidics has inherent microfluidic benefits along with the performances of lateral flow strips that have been proven to be a novel platform for such studies. Multiple marker detection for acute myocardial infarction is necessary for its proper diagnosis. Paper microfluidics-based chemiluminescence techniques can perform multiplexed analysis of antibodies like cardiac troponin I (cTnI), heart-type fatty acid-binding protein (H-FABP), and copeptin [217]. Such devices with a photomultiplier tube (PMT) have great potential for point-of-care analysis. Other techniques like radioimmunoassay, enzyme-linked immunosorbent assay, fluorescence, electrochemistry, and electrochemiluminescence (ECL) can also be used on a chip for diagnostic studies. Early myocardial ischemia stages could be detected with biomarkers such as glycogen phosphorylase isoenzyme BB (GPBB) with a paper microfluidic platform, which is difficult to perform by a conventional method. On the same platform, late markers like cardiac troponin T (cTnT) and CK-MB were also detected [218]. Detailed reviews of microfluidic diagnostic tools can be found elsewhere [219,220,221], as it is not the central part of this work.

## 6. Future Directions

In spite of technological advances in the study of cell biology, cellular behavior, and the pathophysiology of MI, there are many of unconnected questions that have to be solved in order to achieve a more efficient and effective stem cell-based therapy for MI. Understandably, no single therapeutic approach will suit all, but to do what can be done to narrow down stem cells, delivery approaches, and biomaterials to the minimal and most efficient formation is the major challenge requiring a detailed and thorough exploration. However, it is still not clear which stem cell is ideal for treating acute or chronic MI. Which mode of delivery is best or better than others, and which biomaterial is most suitable for stem cell delivery to the infarcted region? Many clinical trials have been completed so far, and many will be conducted in the upcoming years. However, a clear strategy has to be evolved in order to find the best combination of stem cells, delivery approach, and biomaterial components to meet the requirement in most cases, if not all.

Microfluidics is another major research area that has helped in the resolution of many complicated questions such as the selection of biomaterials and the mechanisms involved in transplanted stem cell-mediated MI heart regeneration, as well as the basic pathophysiology of MI. Most importantly, microfluidics can be used to more efficiently understand the angiogenesis process and the therapeutic interventions required to increase the vasculature density in the infarcted region. Eventually, it could help to improve the functionality of the infarcted heart.

## 7. Conclusions

Stem cell delivery to the infarcted myocardium has become indispensable for regenerating the heart so as to regain functionality similar to native cardiac function. To this end, various stem cells have been used for delivery to the infarcted heart. Processes used include direct injection into the myocardium and using a 3D biocompatible scaffold (commonly called a patch) to deliver a higher dose of cells with uniform distribution across the infarcted region. Several mechanisms have been documented where either the transplanted stem cells differentiate into functional CMs and integrate with the heart to improve cardiac function or act through a paracrine manner to induce the regeneration of CMs, increase neovascularization, reduce scar formation, increase ventricular function and decrease remodeling of the myocardium. Hence, the forthcoming mechanistic studies on exosomes and direct tagging of stem cells for monitoring the homing studies after transplantation may further result in translating bench studies to the bedside. 

## Figures and Tables

**Figure 1 cells-10-02538-f001:**
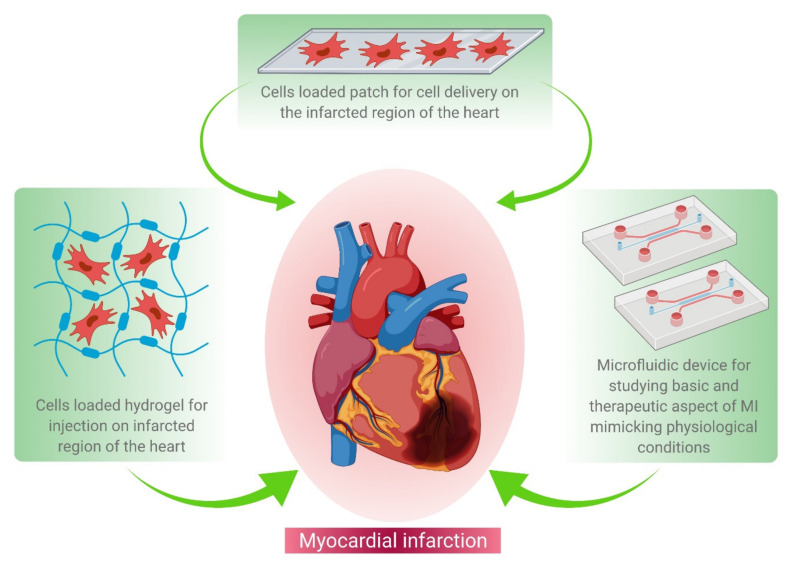
Schematic representation of various tissue engineering approaches for MI treatment. These approaches include hydrogel-based cell delivery (left hand corner), patch-based cell delivery (middle panel), and microfluidics-based drug screening (right corner) during the regenerative therapy of damaged heart tissue.

**Figure 2 cells-10-02538-f002:**
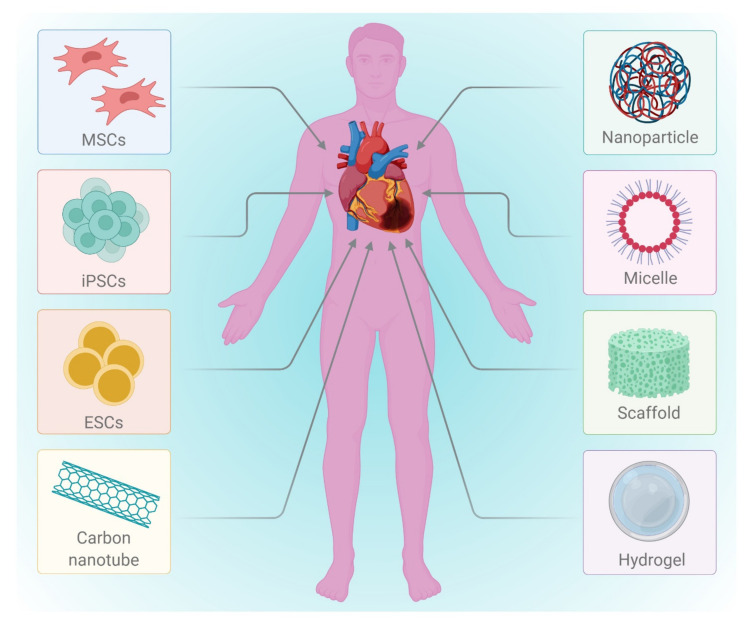
Illustration of stem cell-based regenerative medicine in MI treatment. Various stem cells like MSCs, ESCs, and iPSCs have been employed in MI treatment. Stem cells are delivered through engineered novel biomaterials (via encapsulation) that mimic the native niche. The stem cell-based regenerative therapy is beneficial to either replace the injured area or the whole organ.

**Figure 3 cells-10-02538-f003:**
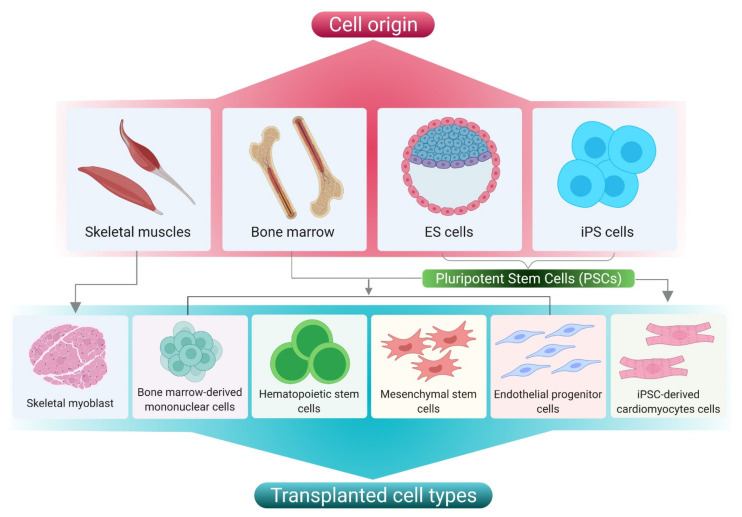
Distinct cells of different origins which are used in the regenerative medicine depicted herein. Depending on the pluripotency, the transplanted cell types can differentiate into various other cells such as skeletal myoblasts, endothelial progenitor cells, chondrocytes, adipocytes, or cardiomyocytes.

**Figure 4 cells-10-02538-f004:**
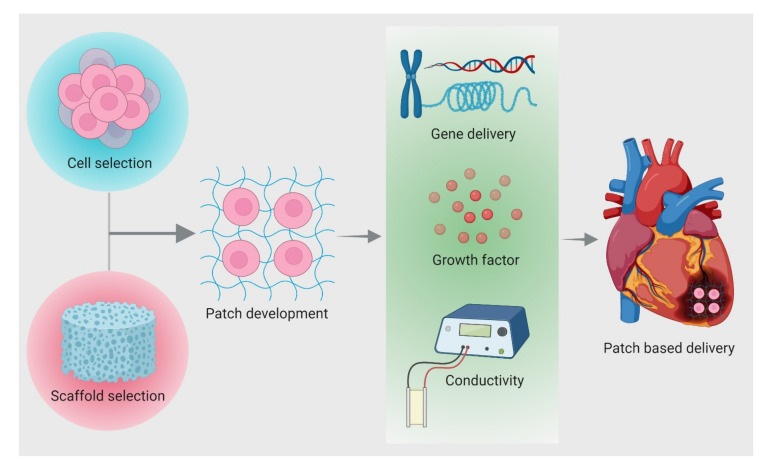
Patch-based cell therapy development started with the fabrication of different patches made up of carbon nanostructures, conductive or non-conductive polymers, hydrogel, etc. followed by patch-based stem cell delivery for MI disease treatment.

**Figure 5 cells-10-02538-f005:**
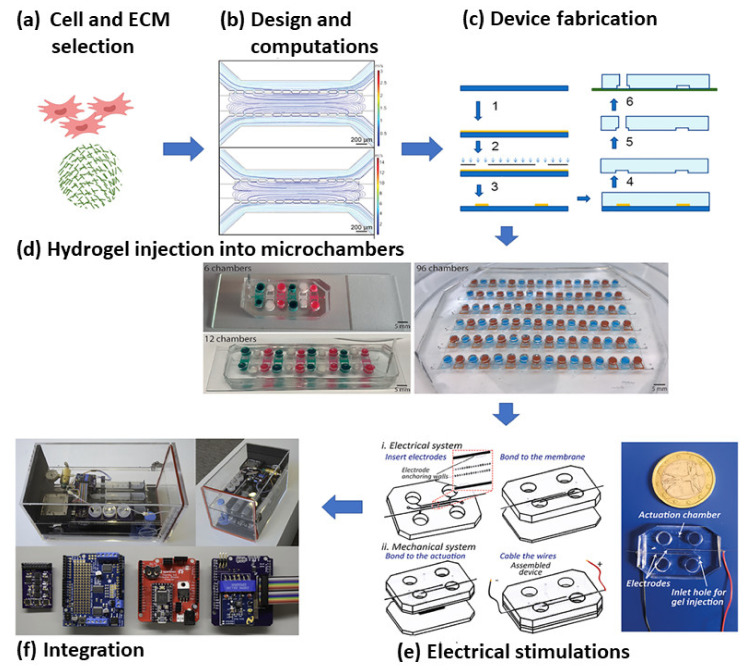
Steps in microfluidic cardiac model generation—(**a**) Selection of cells and extracellular matrix are performed based on the physiology to be studied; (**b**) this is followed by design of the chip through different computational software to achieve the desired flow contours; (**c**) based on the design, the device is fabricated by various microfabrication procedures (the most common being photolithography; the steps of which are 1. spin coating of clean silicon wafer, 2. UV exposure with a mask, 3. dissolution of unwanted resist with developer solution to generate the master pattern, 4. PDMS mold creation from the master pattern, 5. punching of required inlet and outlet holes in the PDMS mold, and 6. bonding of the PDMS mold with a glass plate or another PDMS slab to close the device); (**d**) introduction of cell-laden hydrogel into the device for 3D culture (its selection is based on the mechanical properties needed for the micro tissue under study; (**e**) completion of the electrical circuit required for stimulation of the cardiac cells; (**f**) integration of the device with external circuitry and pumping mechanism for seamless operation of the chip [176,177,178,179].

**Table 2 cells-10-02538-t002:** Types of natural and synthetic polymers used during cardiac tissue engineering.

Natural Polymers	
Chitosan	Hyaluronic acid
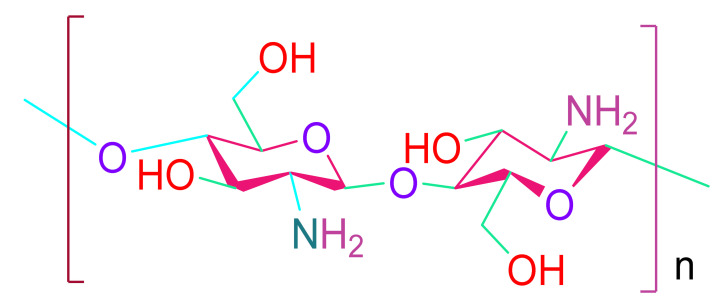	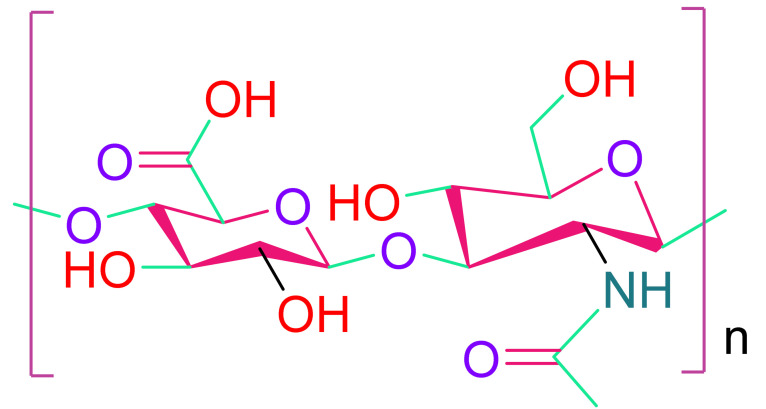
Alginate	Fibrin
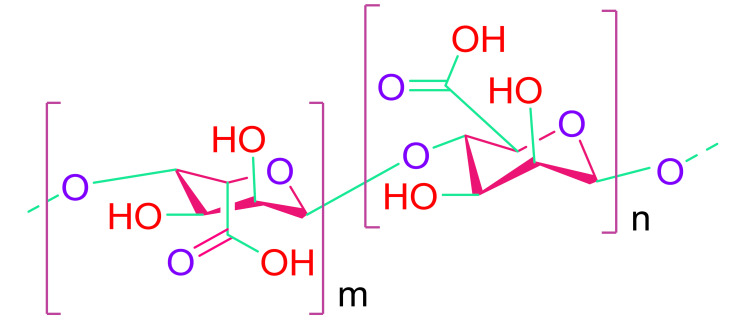	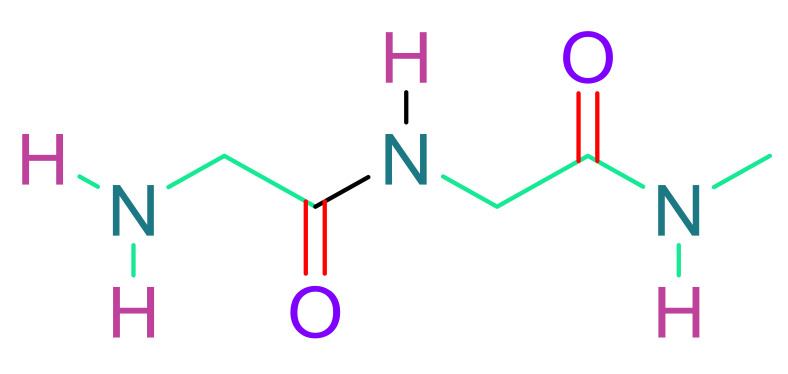
**Synthetic Polymers**	
Poly(glycolic acid)	Poly(ε-caprolactone)
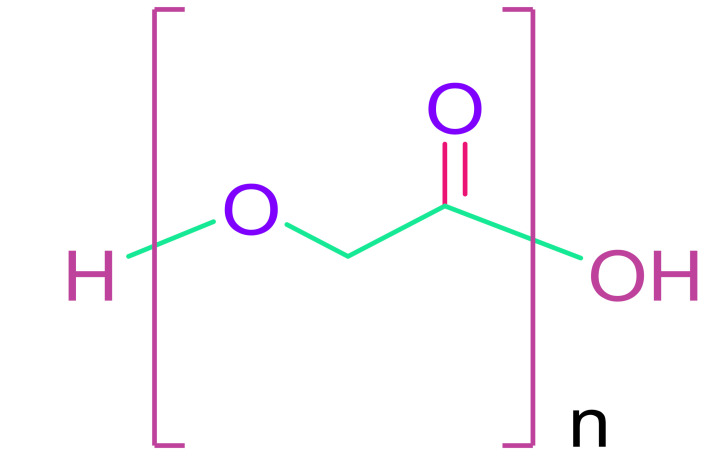	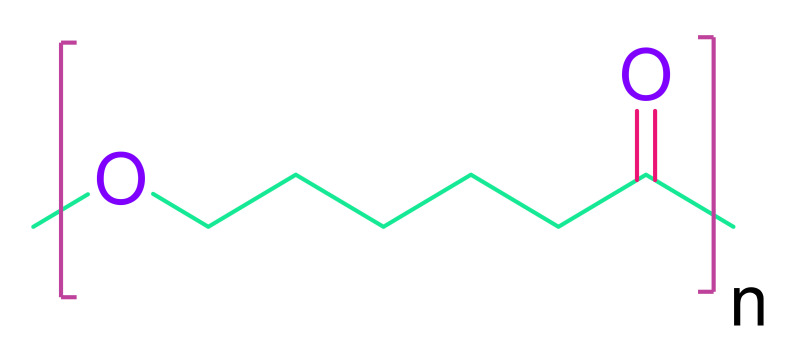
Poly(N-isopropylacrylamide)	Poly(ethylene glycol)
	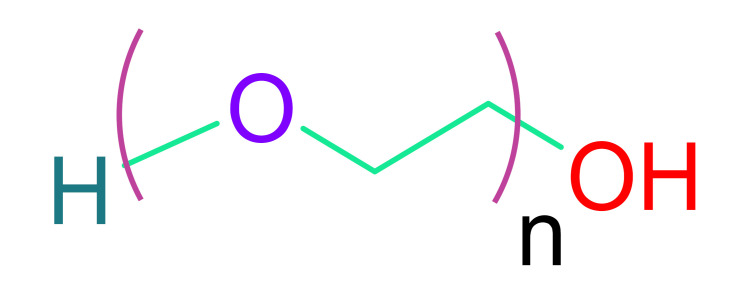
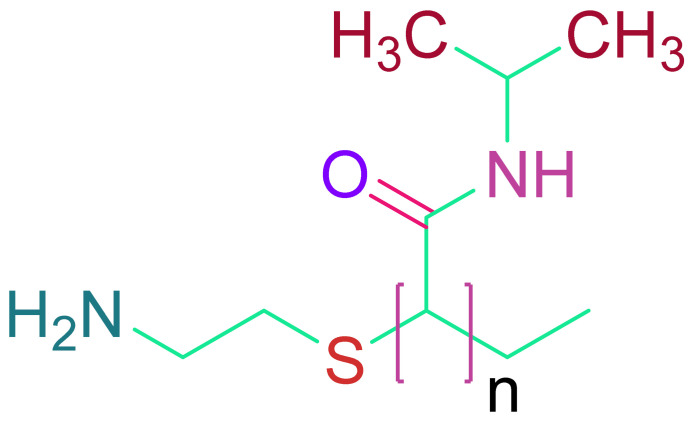	Poly(lactic acid)
	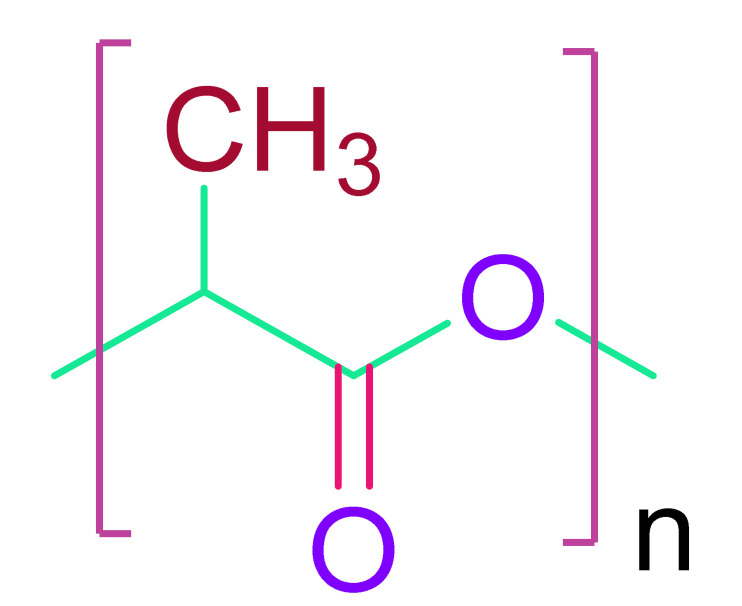
Gelatin methacryloyl 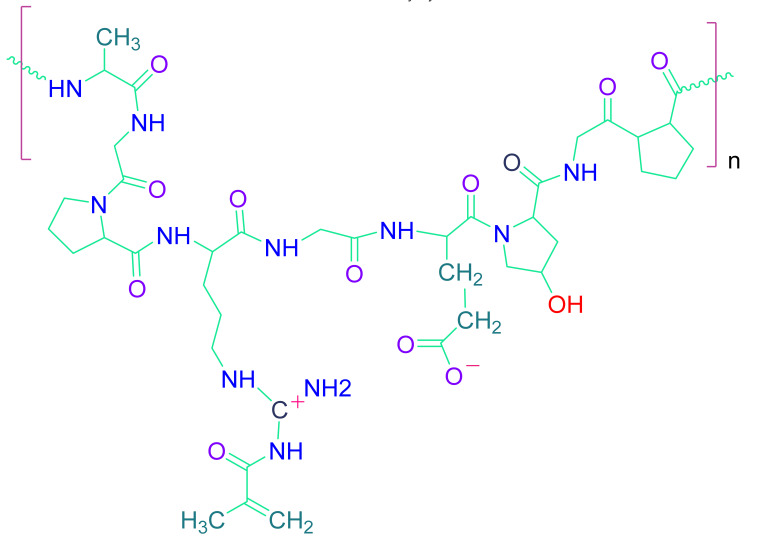	

## Data Availability

Not Applicable.

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
