# Peer review of "Recent Advances in Cardiac Tissue Engineering for the Management of Myocardium Infarction"

_cells, 2021, doi:10.3390/cells10102538_

Round 1
Reviewer 1 Report
This is a well-written and informative overview of the tissue engineering field. However, the review is focused more on the biomaterial part as compared to the stem cell-based cellular therapy part.
I have only some minor comments.
The authors cited two manuscripts from the Anversas group. However, sofar as I know more than 30 publications were retracted. Therefore please check whether the cited manuscripts are retracted or not; Retracted manuscripts should be not cited.
There are also some references that should be cited if applicable:
1) DOI: 10.1186/s13287-020-02089-5 2) Acta Biomater: 2019 Apr 15;89:180-192. doi: 10.1016/j.actbio.2019.03.017. Epub 2019 Mar 9.
Author Response
This is a well-written and informative overview of the tissue engineering field. However, the review is focused more on the biomaterial part as compared to the stem cell-based cellular therapy part.
I have only some minor comments.
The authors cited two manuscripts from the Anversas group. However, so far as I know more than 30 publications were retracted. Therefore, please check whether the cited manuscripts are retracted or not; Retracted manuscripts should be not cited.
Author Response:We are thankful to the reviewer for providing the important suggestions and comments on our manuscript. As suggested by the reviewer, we checked the manuscripts published by Anversas group and found that the cited papers are not retracted.
There are also some references that should be cited if applicable:
1) DOI: 10.1186/s13287-020-02089-5 2) Acta Biomater: 2019 Apr 15;89:180-192. doi: 10.1016/j.actbio.2019.03.017. Epub 2019 Mar 9.
Author Response: As suggested by the reviewer, we have cited the relevant reference in the revised manuscript. (Line no. 509)
Reviewer 2 Report
The authors have assembled a detailed and rather extensive review in the area of cardiac tissue engineering in the context of addressing myocardial infarction. The review is deserving of publication in this journal after addressing the following concerns:
In the section regarding cell therapy for MI, it should be mentioned that aggregate injection (not single cells) is another strategy that is a like middle ground between single cell and patch transplantation. I recommend citing the following papers in support:
1) https://www.nature.com/articles/s41467-020-17742-z?elqTrackId=95b3266057564f9e9a5f22a4edf97718
2) https://www.ncbi.nlm.nih.gov/pmc/articles/PMC5095349/
3) https://pubs.acs.org/doi/10.1021/acsnano.7b01008
4) https://journals.plos.org/plosone/article?id=10.1371/journal.pone.0050491
5) https://pubmed.ncbi.nlm.nih.gov/21251587/
While microfluidcs have and will provide great insight into MI biology and therapies, there are other cardiac tissue engineered platforms that are directly related to this manuscript that should be mentioned either in the review or in future outlook section, such as the following papers:
1) https://www.nature.com/articles/s41551-020-0539-4
2) https://www.liebertpub.com/doi/abs/10.1089/ten.tea.2018.0212
3) https://www.sciencedirect.com/science/article/pii/S193152441930194X
Author Response
The authors have assembled a detailed and rather extensive review in the area of cardiac tissue engineering in the context of addressing myocardial infarction. The review is deserving of publication in this journal after addressing the following concerns:
In the section regarding cell therapy for MI, it should be mentioned that aggregate injection (not single cells) is another strategy that is a like middle ground between single cell and patch transplantation. I recommend citing the following papers in support:
1) https://www.nature.com/articles/s41467-020-17742 z?elqTrackId=95b3266057564f9e9a5f22a4edf97718
2) https://www.ncbi.nlm.nih.gov/pmc/articles/PMC5095349/
3) https://pubs.acs.org/doi/10.1021/acsnano.7b01008
4) https://journals.plos.org/plosone/article?id=10.1371/journal.pone.0050491
5) https://pubmed.ncbi.nlm.nih.gov/21251587/
Author Response: We are very grateful to the reviewer for providing the critical suggestions and valuable comments on our manuscript. As suggested by the reviewer we have included a subsection on the cell aggregates or strategy regarding the aggregate injection in the revised manuscript.
3h. Cells Aggregates
Although stem cell transplantation is currently implemented clinically, it is difficult to accomplish minimally invasive injectable cell delivery while retaining high cell retention and animal survival. Strategies involving stem cell retention in the infarct region is being studied such as delivering cell aggregates, and patch-based therapy. Tang et al.,demonstrated the safety and efficacy of encapsulating human cardiac stem cells (hCSCs) in thermosensitive poly (N-isopropylacrylamineco-acrylic acid) or P(NIPAM-AA) nanogel in mouse and pig models of MI. Unlike xenogeneic hCSCs injected in saline, injection of nanogel-encapsulated hCSCs did not elicit systemic inflammation or local T cell infiltration in immunocompetent mice. The developed thermosensitive nanogels can be used as a stem cell carrier: the porous and convoluted inner structure not only allows nutrient, oxygen, and secretion diffusion but also prevents the stem cells from being attacked by immune cells [205]. Compared to the traditional approaches of single cell injection, cell aggregate deliveries have demonstrated higher retention of cells and prevention of teratoma development [206]. Another such study on cell aggregates by Bauer et. al. showed the better survivability of these aggregates could be attributed to the imitation of endogenous state by ensuring adequate cell-cell interaction [207]. Bioengineered 3D framework which enhances cellular contact while still allowing for certain cell ratios was developed by Monsanto et al.,These injectable cardio clusters enhance adhesions and reduces cell loss [208].(Line no. 268 – 283)
While microfluidics has and will provide great insight into MI biology and therapies, there are other cardiac tissue engineered platforms that are directly related to this manuscript that should be mentioned either in the review or in future outlook section, such as the following papers:
1) https://www.nature.com/articles/s41551-020-0539-4
2) https://www.liebertpub.com/doi/abs/10.1089/ten.tea.2018.0212
3) https://www.sciencedirect.com/science/article/pii/S193152441930194X
Author Response: We thank the reviewer for the suggestion. The above-mentioned references have been cited in the revised manuscript and the importance of the work has been cited.
In conventional culture conditions there are few models which could capture the physiology of infarcted heart tissue using cardiac organoids that mimic the oxygen gradient and adrenergic stimulus in MI [214].(Line no. 791-793)
It is difficult to study IRI in human models, although there are in vitromodels which could recapitulate the oxygen concentration and pH condition in IRI [215]. Such models also show that cardioprotective therapeutics and ischemic preconditioning have the ability to reduce IRI [216].(Line no. 832-835)
Reviewer 3 Report
In the article Recent Advances in Cardiac Tissue Engineering for the Management of Myocardium Infarction Sharma et al. aim to review tissue engineering based strategies to i) repair the injured heart and ii) their application in myocardial infarction research.
The first part on cardiac repair is dived in a part that focuses on the different cell types that have been applied to repair the heart and a second part on the engineering strategies that have been used to generate engineered myocardium. I have substantial concerns about the first part.
The authors miss major developments within the field over the last ten years. Additionally, there are quite some errors, e.g. the authors write that “Cardiomyocytes, smooth muscle cells, and endothelial cells can all be differentiated from cardiovascular lineage specific progenitor cells resides in the heart. Injection after MI demonstrated in vivo regenerative capacity, but progenitor cells were rejected by the host.“ Yet, in contrast to their statement it is well established that there is no resident cardiac progenitor cell type (Li et al. Genetic Lineage Tracing of Nonmyocyte Population by Dual Recombinases, Circulation 2018). Unfortunately, there many more similar mistakes. For example the authors write that “Chong et al. were the first ones to obtain a large number of cardiomyocytes from ESCs and used them to repair injured myocardium.” It is true that Chong et al. were the first to publish a non-human primate model for ESC-CMs based repair. However, this work was based on several other studies from the same (and other) groups that are not mentioned by the authors. The authors also miss the most relevant publication in the field of tissue engineered heart repair from the laboratories of Jay Zhang, Wolfram Zimmermann, Nenad Bursac and Thomas Eschenhagen. Overall, this first part does not reflect the current knowledge on tissue engineered heart repair.
The second more technical part provides an extensive overview on biomaterials that have been used for engineer myocardium. I only have minor comments on this part. It is well structured but mainly lists a variety of studies which makes it difficult to read and a more critical appraisal would be helpful.
The first part of the article is followed by a second part on microfluidics in myocardial infarction research and drug discovery and disease modelling. Eventually the authors conclude their article with a third short paragraph on point of care devices. In my opinion this is too much to be covered in one review. The work remains rather superficial and does not provide any background information. In this regard I do not see how it can help readers that are new the field of cardiac regeneration but also does not provide any additional benefit for scientists working on cardiac repair.
Author Response
In the article Recent Advances in Cardiac Tissue Engineering for the Management of Myocardium Infarction Sharma et al. aim to review tissue engineering based strategies to i) repair the injured heart and ii) their application in myocardial infarction research.
The first part on cardiac repair is dived in a part that focuses on the different cell types that have been applied to repair the heart and a second part on the engineering strategies that have been used to generate engineered myocardium. I have substantial concerns about the first part.
The authors miss major developments within the field over the last ten years. Additionally, there are quite some errors, e.g. the authors write that “Cardiomyocytes, smooth muscle cells, and endothelial cells can all be differentiated from cardiovascular lineage specific progenitor cells resides in the heart. Injection after MI demonstrated in vivo regenerative capacity, but progenitor cells were rejected by the host.“ Yet, in contrast to their statement it is well established that there is no resident cardiac progenitor cell type (Li et al. Genetic Lineage Tracing of Nonmyocyte Population by Dual Recombinases, Circulation 2018).
Author Response: We are thankful to the reviewer for providing the valuable comments on our manuscript. We agree with the reviewer’s comment and we have revised the manuscript accordingly.
We state:
Lineage tracing studies without a specific cardiac marker showed the existence of endogenous CSCs in the fetal heart; however, it was also pointed out in the study that lack of data support the existence of CSCs in the adult heart respectively. Moreover, recent studies revealed progenitors supporting damaged heart regeneration viasecreting factors that rejuvenate the resident CSCs to counter balance the lost cells. However, the massive damage required a high number of cells to maintain the homeostasis. Recent clinical studies conducted by the CADUCEUS, using autologous cardiosphere derived cells (CDCs), showed improved heart function [44-46].(Line no. 234-240)
Unfortunately, there many more similar mistakes. For example the authors write that “Chong et al. were the first ones to obtain a large number of cardiomyocytes from ESCs and used them to repair injured myocardium.” It is true that Chong et al. were the first to publish a non-human primate model for ESC-CMs based repair. However, this work was based on several other studies from the same (and other) groups that are not mentioned by the authors. The authors also miss the most relevant publication in the field of tissue engineered heart repair from the laboratories of Jay Zhang, Wolfram Zimmermann, Nenad Bursac and Thomas Eschenhagen. Overall, this first part does not reflect the current knowledge on tissue engineered heart repair.
Author Response: As per the reviewer’s suggestions, the pioneer studies have been included in the revised manuscript.
Few pioneer studies such as engineering of sheet-based cardiac patches constructed to harbor well aligned and interconnected cardiomyocytes for successful implants to regenerate the myocardium [220]. Zimmermann et al.,constructed myocardial tissue employing 3D models to mimic the native heart muscles which had resulted in the restoration and improvement of cardiac function. Helfer & Bursac, demonstrated a versatile framed hydrogel methodology to generate engineered cardiac tissue with enhanced mature functional properties. Therefore, it is well understood that in order to engineer a cardiac patch, the most important prerequisite is the cells that proliferate and gain functionality in the infarcted region. Different strategies involved in patch design for treating injured myocardium are shown in Figure 4and described in the following section.(Line no. 288-296)
The second more technical part provides an extensive overview on biomaterials that have been used for engineer myocardium. I only have minor comments on this part. It is well structured but mainly lists a variety of studies which makes it difficult to read and a more critical appraisal would be helpful.
Author Response:As per the reviewer’s suggestion, the manuscript has been revised as the narrative and the lucid flow of the manuscript in order to attract more readers.
The first part of the article is followed by a second part on microfluidics in myocardial infarction research and drug discovery and disease modelling. Eventually the authors conclude their article with a third short paragraph on point of care devices. In my opinion this is too much to be covered in one review. The work remains rather superficial and does not provide any background information. In this regard I do not see how it can help readers that are new the field of cardiac regeneration but also does not provide any additional benefit for scientists working on cardiac repair.
Author Response:We thank the reviewer for this valuable comment. As mentioned in the manuscript disease diagnosis is an important aspect of cardiac tissue engineering. It is required to understand the cause of myocardial infraction and the stage of the injury in order to plan a repair strategy for the cardiac tissue. Point of care devices also play a vital role during the treatment process of a patient. Though it is an important part of cardiac tissue engineering the core aspect of the manuscript is not on disease diagnosis as put forward by the reviewer. Because of this reason this part is not covered extensively. The revised manuscript has been modified to reflect this idea to the reader.